



# Results from a Wake Steering Experiment at a Commercial Wind Plant: Investigating the Wind Speed Dependence of Wake Steering Performance

Eric Simley[1], Paul Fleming[1], Nicolas Girard[2], Lucas Alloin[3], Emma Godefroy[3], and Thomas Duc[3]

[1]National Wind Technology Center, National Renewable Energy Laboratory, Golden, CO, 80401, USA
[2]ENGIE Digital, 59 rue Denuzière, 69002 Lyon, France
[3]ENGIE Green France, 59 rue Denuzière, 69002 Lyon, France

**Correspondence:** Eric Simley (eric.simley@nrel.gov)

**Abstract.** Wake steering is a wind farm control strategy in which upstream wind turbines are misaligned with the wind to redirect their wakes away from downstream turbines, thereby increasing the net wind plant power production and reducing fatigue loads generated by wake turbulence. In this paper, we present results from a wake steering experiment at a commercial wind plant involving two wind turbines spaced 3.7 rotor diameters apart. During the three-month experiment period, we estimate that

wake steering reduced wake losses by 5.7% for the wind direction sector investigated. After applying a long-term correction based on the site wind rose, the reduction in wake losses increases to 9.8%. As a function of wind speed, we find large energy improvements near cut-in wind speed, where wake steering can prevent the downstream wind turbine from shutting down. Yet for wind speeds between 6–8 m/s, we observe little change in performance with wake steering. However, wake steering was found to improve energy production significantly for below-rated wind speeds from 8–12 m/s. By measuring the relation-

ship between yaw misalignment and power production using a nacelle lidar, we attribute much of the improvement in wake steering performance at higher wind speeds to a significant reduction in the power loss of the upstream turbine as wind speed increases. Additionally, we find higher wind direction variability at lower wind speeds, which contributes to poor performance in the 6–8-m/s wind speed bin because of slow yaw controller dynamics. Further, we compare the measured performance of wake steering to predictions using the FLORIS (FLOw Redirection and Induction in Steady State) wind farm control tool

coupled with a wind direction variability model. Although the achieved yaw offsets at the upstream wind turbine fall short of the intended yaw offsets, we find that they are predicted well by the wind direction variability model. When incorporating the predicted achieved yaw offsets, estimates of the energy improvement from wake steering using FLORIS closely match the experimental results.

## 1 Introduction

Wake steering is a wind farm control strategy for mitigating wake effects in which upstream wind turbines are misaligned with the wind, thereby deflecting their wakes away from downstream turbines (Dahlberg and Medici, 2003; Wagenaar et al., 2012; Boersma et al., 2017). Despite the power loss from yaw misalignment, wake steering can increase the net power produced by



the wind plant because of the higher wind speeds encountered by downstream wind turbines. Additionally, research suggests that wake steering can reduce fatigue loads on downstream turbines by redirecting high-turbulence wake flow away from the

turbines and avoiding partial wake interactions which can cause asymmetric rotor loading (Kanev et al., 2018; López et al., 2020).

The potential for wake steering to improve wind plant power production has been demonstrated for stationary wind conditions using high-fidelity computational fluid dynamics (CFD) simulations, engineering models, and wind tunnel experiments. Using the National Renewable Energy Laboratory's (NREL's) Simulator fOr Wind Farm Applications (SOWFA) large-eddy

simulation (LES) tool, Gebraad et al. (2016) observed a 13% increase in power production for a $2 \times 3$ array of wind turbines with 5-rotor-diameter ($5D$) spacing in neutral atmospheric conditions. Vollmer et al. (2016) used LES to investigate the impact of atmospheric stability on wake steering, finding that the ability to control the wake position strongly depends on stability; low-turbulence stable atmospheric conditions were shown to be more favorable for wake steering than unstable conditions with higher turbulence. Although high-fidelity CFD simulations are valuable for studying the physics of wake steering, computa-

tionally efficient engineering models are needed to optimize wake steering controllers and to estimate performance for a variety of wind conditions. Gebraad et al. (2017) and King et al. (2021) used NREL's FLOw Redirection and Induction in Steady State (FLORIS) engineering wind farm control tool (NREL, 2021) to estimate annual energy production improvements of 3.8% for a 60-turbine wind plant and 2.8% for a wind plant with 38 wind turbines, respectively. Using scaled wind turbine models, wind tunnel experiments have been used to investigate the effectiveness of wake steering beyond simulation environments. For

example, using two-turbine arrays, Adaramola and Krogstad (2011) and Campagnolo et al. (2016) achieved net power gains of 12% for a $3D$ turbine spacing and 21% for a $4D$ spacing, respectively. Similarly, Bastankhah and Porté-Agel (2019) measured a 17% increase in power production for a row of five model wind turbines spaced $5D$ apart.

To bridge the gap between simulations and wind tunnel experiments with static wind conditions and successful implementation of wake steering in the field, several recent studies have investigated the design of wake steering controllers for realistic

dynamic wind conditions. Bossanyi (2018) used field measurements of wind conditions as inputs to a dynamic engineering wind farm control model to evaluate combined yaw and power set point control for increasing energy production and reducing loads. Rather than relying on the turbines' existing yaw controllers to implement the intended yaw offsets, Bossanyi (2018) found that directly yawing the turbines at regular time intervals improved the controller performance in dynamic wind conditions. By modifying the FLORIS wind farm control tool to model dynamic wind conditions, Kanev (2020) optimized the

parameters of a simple yaw offset lookup table-based wake steering controller. A good balance between energy production and the required yaw actuation was achieved by (1) updating the yaw offset command at least every 2 minutes, (2) filtering the wind direction input using a time constant similar to the update rate, and (3) including hysteresis on the yaw offset command to reduce yaw activity. The concept of robust wake steering control has been explored by several authors to address the challenge of operating in dynamic wind conditions (Rott et al., 2018; Simley et al., 2020b; Quick et al., 2020). Specifically, realizing

that the wind direction can vary considerably while a turbine's yaw position remains fixed, the authors identified yaw offsets that maximize energy production assuming a certain degree of wind direction uncertainty. As a last example, Doekemeijer et al. (2020) presented a closed-loop wake steering controller that incorporates measurements from multiple wind turbines to





estimate wind plant-level wind conditions, updating the yaw offsets accordingly. Using CFD simulations with time-varying mean wind directions across the wind plant, the authors demonstrated a 1.4% increase in energy production for a six-turbine array when updating the turbines' yaw positions every 20 seconds.

Following an early inconclusive test of wake steering discussed by Wagenaar et al. (2012), recently several wake steering experiments at commercial wind plants have been described in the literature. Fleming et al. (2017) implemented wake steering control on a single wind turbine in an offshore wind plant in China to benefit three turbines located $7D$, $8.6D$, and $14.3D$ downstream in different directions. The authors reported power gains as high as 29% for certain wind directions for the $7D$ spacing, but they highlighted large uncertainty in the results because of a lack of data. An extensive field campaign at a U.S. land-based wind plant, in which two turbines were controlled to redirect their wakes away from a turbine $2.9D$ and $5D$ downstream for southerly and northerly wind directions, respectively, is documented by Fleming et al. (2019, 2020). Limiting the controller to clockwise yaw misalignments relative to the wind direction, the authors showed a 6.5% reduction in overall wake losses from wake steering for both turbine combinations. Further, the authors observed that wake steering is significantly more effective during nighttime or stable atmospheric conditions than during the daytime or in unstable conditions. Howland et al. (2019) implemented wake steering on a row of six wind turbines spaced $3.5D$ apart at a wind plant in Alberta, Canada. Using a fixed yaw offset of $20°$ for the first five turbines encountering the wind, the authors measured power gains of up to 47% and 13% for wind speeds between 5–6 m/s and 7–8 m/s, respectively. Large increases in power—as well as significant reductions in the variability of the power production—were achieved at low wind speeds because wake steering caused the waked turbines to shut down less frequently by maintaining wind speeds above the cut-in speed. Last, Doekemeijer et al. (2021) described a wake steering experiment at a wind plant in Italy in which two turbines were controlled to improve the net power production for either a row of three turbines or pairs of turbines spaced $5.2D$ to $6.5D$ apart, depending on the wind direction. Using both positive and negative yaw offsets, the authors observed increases in energy production of up to 35% for the two-turbine scenario and 16% for the row of three turbines while also acknowledging net *losses* in energy production for certain wind directions. Additionally, in some cases the authors measured unexpected gains in energy production for the misaligned turbines, suggesting uncertainty in the results.

In this paper, we present results from a wake steering campaign at a land-based wind plant in France owned and operated by ENGIE Green in which a single wind turbine is controlled to increase the power production of a second turbine $3.7D$ downstream. Similar to work by Fleming et al. (2019, 2020), the performance of wake steering control is analyzed in terms of the impact on energy production for the pair of turbines as well as the ability of the upstream turbine to achieve the desired yaw offsets. Moreover, a forward-facing nacelle lidar installed on the upstream turbine is used to measure the yaw misalignment and inflow wind speed to help assess wake steering performance. The main contributions of the paper are as follows. First, we highlight the wind speed dependence of the energy improvements as well as the ability to achieve the intended yaw offsets, from cut-in to nearly rated wind speed. Next, we compare the change in energy production and the achieved offsets to model predictions based on the FLORIS wind farm control tool accounting for realistic wind direction variability. To determine whether the nacelle wind vane can be reliably used to implement wake steering, we compare the yaw misalignments measured by the wind vane to those measured by the nacelle lidar; we then suggest wind speed-dependent corrections to the wind vane to





more accurately measure the true yaw misalignment. Finally, we use measurements of yaw misalignment and wind speed from the nacelle lidar to determine the relationship between yaw misalignment and power production as a function of wind speed.

The rest of the paper is organized as follows. Section 2 provides an overview of the field experiment, including the wind plant and turbine specifications, instrumentation, control strategy, and wind resource information. The FLORIS wind farm control model and the wind direction variability model used to predict wake steering performance are presented in Section 3. Data processing steps performed before analyzing wake steering performance are described in Section 4. Section 5 compares the predicted and achieved yaw offsets during the experiment as a function of wind direction and wind speed. Suggested corrections

to the wind vane measurements, estimated using nacelle lidar measurements, are described in Section 5.3. Next, the wind speed-dependent relationship between yaw misalignment and power production for the upstream turbine is investigated in Section 6, again using nacelle lidar measurements. Section 7 presents the impact of wake steering on overall energy production for the two wind turbines as well as the change in energy as a function of wind speed. Results are compared to FLORIS predictions to help validate the FLORIS model. Last, Section 8 concludes the paper with a discussion of the results and suggestions for

further research.

## 2   Field Experiment Overview

The wind plant used for the experiment is Sole du Moulin Vieux (SMV), a commercial wind plant operated by ENGIE Green. It is located in the northern part of France, approximately midway between Paris and Lille, and was already used in previous field tests as part of the SMARTEOLE project (Ahmad et al., 2017; Duc et al., 2019). It consists of seven Senvion MM82 wind

turbines (rotor diameter of $D$ = 82 m, nominal power of 2050 kW, hub height of 80 m) organized in a north-south axis, as shown by the layout in Fig. 1. The terrain is simple, but a small forest south of the plant slightly disturbs the flow for southerly winds.

Only two turbines, SMV5 and SMV6, are considered for the wake steering experiment. They were chosen because of the short spacing between them ($3.7D$) and their alignment close to prevailing wind directions observed at the site, as shown by

the long-term wind rose in Fig. 2. Consequently, SMV5 experiences a strong and frequent wake from SMV6, which makes it a very interesting case for testing wind farm control strategies. The guaranteed power and thrust curves for the Senvion MM82 wind turbines are shown in Fig. 3.

### 2.1   Instrumentation

Some additional instrumentation was set up on the wind plant for the wake steering experiment. First, all turbines were equipped

with a supervisory control and data acquisition (SCADA) system allowing 1-Hz data for the most critical variables to be recorded. A Vaisala Triton sodar was installed in the proximity of turbines SMV5 and SMV6, and a Leosphere WindCube v1 profiling lidar was installed between turbines SMV2 and SMV3. Their precise locations are presented in Fig. 1. Although data from the sodar and profiling lidar were not used extensively during the analysis, they were used to cross-check and



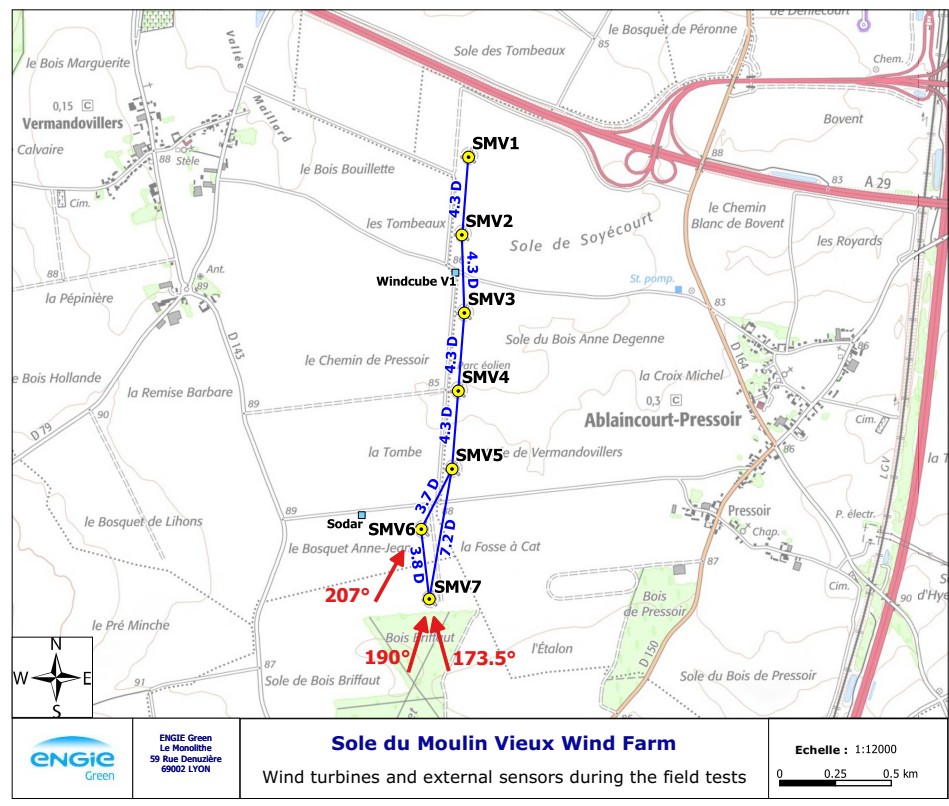

**Figure 1.** Layout of the SMV wind plant and experimental setup for the wake steering field experiment. Along with the WindCube v1 lidar and the Triton sodar indicated on the map, a WindCube Nacelle lidar and a GNSS compass were installed on top of SMV6. Distances between wind turbines (normalized by the rotor diameter $D = 82$ m) and directions related to SMV6 and SMV5 are also shown.

validate wind measurements from the turbines and to identify the best references for assessing the ambient wind conditions.
Additionally, measurements from the WindCube lidar were used to estimate the turbulence intensity distribution at the site.

A WindCube Nacelle lidar was installed on top of the controlled turbine, SMV6, to measure the wind inflow, including the turbine's yaw misalignment with respect to the incoming wind direction. This sensor collects radial wind speed values from four beams at a sample frequency of 4 Hz at 10 range gates spanning 50–200 m upstream. For the lidar-based analyses presented here, we use the estimated horizontal wind speeds and wind directions at hub height provided by the lidar. We use
measurements at a range of 150 m ($1.8D$) to determine wind speed and the average of measurements at 100 m, 150 m, and 200 m to estimate yaw misalignment.

Additionally, a Hemisphere GNSS compass was fixed on the nacelle of SMV6 to monitor the offset of the turbine's reported yaw orientation relative to north during the experiment. Indeed, this offset is known to deviate with time (van der Hoek et al., 2019) and must be calibrated properly when realizing wind farm control experiments. Figure 4 displays the evolution of
this north offset with time by looking at the difference between SMV6's yaw position signal and the GNSS measurement of



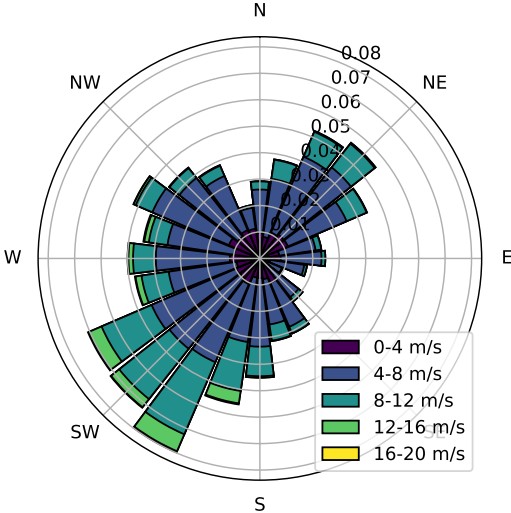

**Figure 2.** Long-term wind rose for the SMV wind plant at hub height (80 m). It was obtained through a correlation process between short-term met-mast measurements on-site and long-term reference wind data (ERA5 reanalysis data).

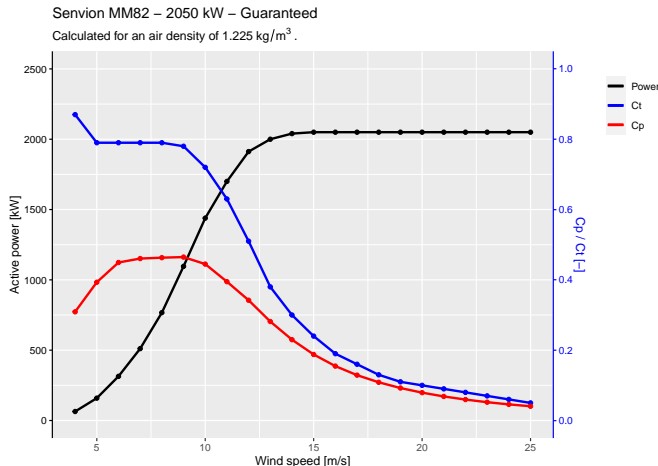

**Figure 3.** Guaranteed power and thrust curves for the Senvion MM82 wind turbines at the SMV wind plant.

the nacelle orientation. It shows that even though some drifts were experienced at the end of 2019, a very stable offset was maintained for the full duration of the field experiment, February 17–May 25, 2020.

Before analyzing the data, all variables are downsampled to 1-minute average values. As will be discussed further in Section 3.2, 1-minute samples are intended to provide a balance between averaging small-scale turbulent variations and distin-

guishing between time-varying wind conditions.

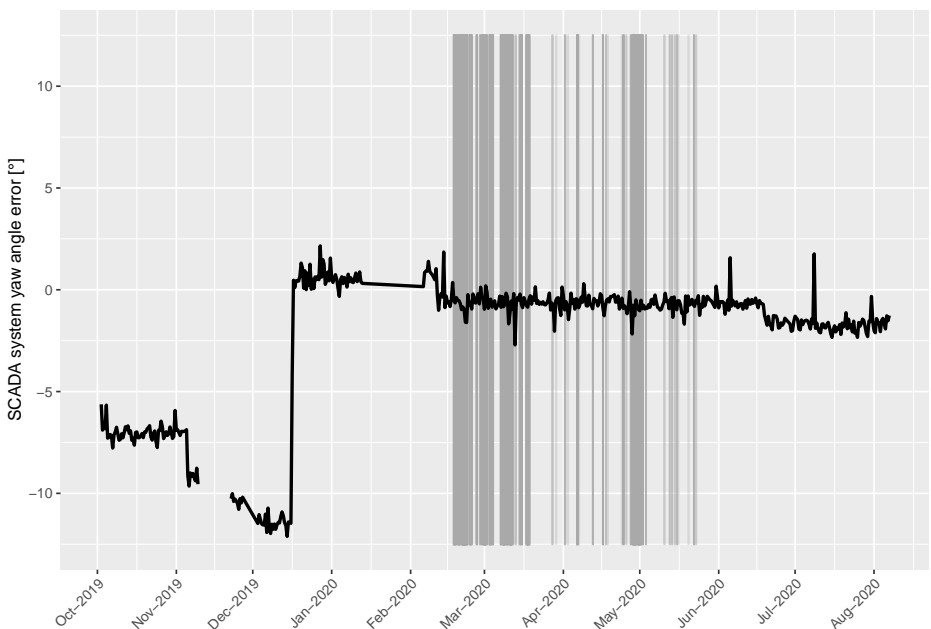

**Figure 4.** Difference between the yaw position of SMV6 measured by the GNSS compass and the turbine SCADA system. The values have been averaged during a period of 12 hours to remove noise associated with the two signals. The vertical gray lines indicate periods when the turbine was intentionally misaligned for the wake steering experiment.

## 2.2 Wake Steering Controller

Because the wind turbine controller could not be accessed or modified for this experiment, the wake steering strategy was implemented following the same approach as in Fleming et al. (2020). A control box was installed to read the incoming relative wind direction signal from the nacelle wind vane installed on the turbine and to apply an offset before sending it to

the turbine's existing yaw controller, thereby inducing the intended yaw offset. The control logic implemented in the control box is illustrated in Fig. 5. As shown in Fig. 5, because the lookup table defining the offset angles is dependent on both wind speed and direction, the nacelle wind speed and yaw position are also used as inputs to the control box. The measured wind direction—formed by combining the absolute yaw position and the relative wind vane direction—and wind speed are passed through low-pass filters with a time constant of 60 seconds before they are used to determine the corresponding target yaw

offset in the lookup table. Finally, a toggle allows the yaw offsets to alternate between the target offsets and zero offset to analyze the effect of wake steering in wind conditions that are similar to the baseline yaw control case. Note that the unfiltered wind speed signal recorded through the control box is compared to the same signal measured by the 1-Hz SCADA system to remove any time lag between the two systems and to ensure that the two clocks are correctly synchronized.

The applied yaw offsets are represented by the offset schedule in Fig. 6. To avoid excessive yaw activity from switching

between large positive and negative values of yaw misalignment, only positive yaw offsets are used in this experiment. Although





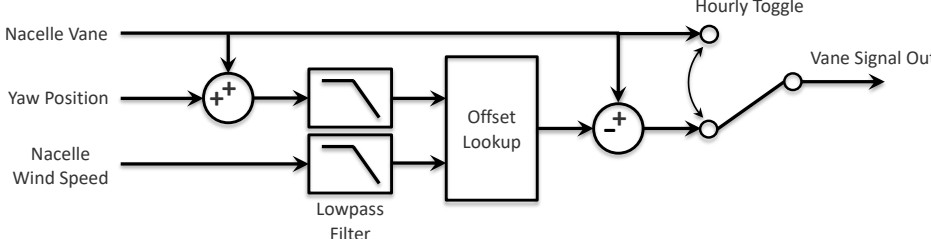

**Figure 5.** Yaw offset control architecture with hourly toggling between baseline and wake steering control. The output wind vane signal is input to the wind turbine's existing yaw controller.

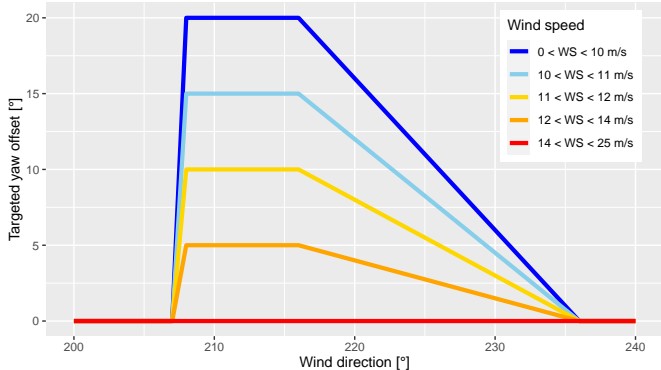

**Figure 6.** Target yaw offset schedule for SMV6 as a function of wind direction and wind speed.

energy improvements from wake steering are expected for both positive and negative yaw offsets, research suggests that, overall, wake steering is more effective with positive yaw misalignment (Fleming et al., 2018; Nouri et al., 2020). Further, results from a field experiment investigating the impact of yaw misalignment on loads show a reduction in blade loads for positive yaw offsets but an increase in loads for negative offsets (Damiani et al., 2018). Adhering to an upper bound of 20°,
imposed to manage structural loads, SMV6 is misaligned by up to 20° for full wake conditions (208°–216°), and then this angle is linearly reduced throughout the partial wake sector until it reaches zero at a wind direction of 236°. The yaw offset schedule for wind speeds below 10 m/s shown in Fig. 6 is a simplified form of the optimal yaw offsets for maximizing the combined power of SMV5 and SMV6 determined using FLORIS. The FLORIS model used when designing the controller was calibrated using data from a previous wake steering experiment performed on the same wind turbines. Additional considerations were
included in the yaw offset schedule to provide some robustness to wind direction uncertainty, following the approach discussed by Simley et al. (2020b); specifically, yaw offsets are applied for a wider sector of wind directions in the partial wake region than suggested using the original FLORIS model. Last, to further reduce the loading at higher wind speeds, the target offsets are reduced in four steps above 10 m/s until wake steering is stopped for wind speeds above 14 m/s.



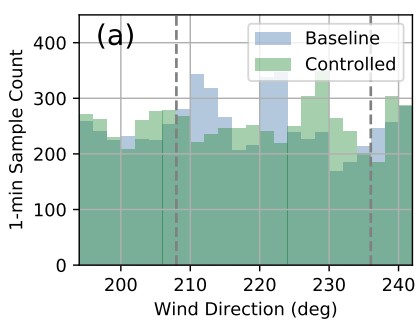
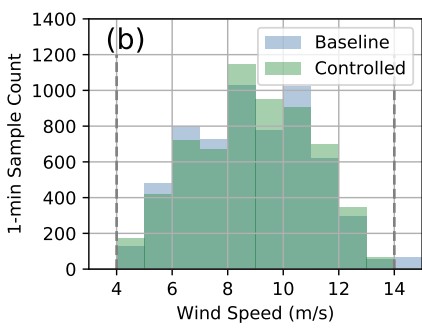
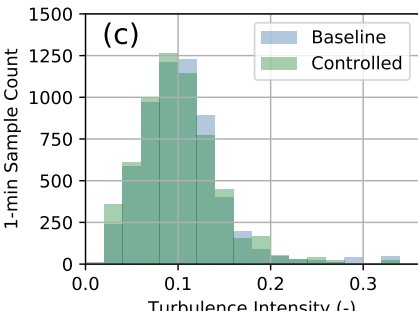

**Figure 7.** Number of 1-minute samples collected for the baseline and the controlled periods grouped by (a) wind direction, (b) wind speed, and (c) turbulence intensity. The wind speed and turbulence intensity distributions contain only periods corresponding to wind directions between 195° and 241°. The gray dashed lines encompass the wind directions and wind speeds where wake steering is intended.

## 2.3 Wind Conditions

The distributions of the wind directions, wind speeds, and turbulence intensities analyzed during the wake steering experiment for the baseline and the controlled periods are shown in Fig. 7 for the wind direction sector between 195° and 241° investigated in this paper. Specifically, Fig. 7 shows the number of 1-minute samples obtained for each wind direction, wind speed, and turbulence intensity bin. Note that the reference wind direction and wind speed measurements are obtained from nacelle-based wind turbine sensors, as will be discussed in Section 4. Turbulence intensity is estimated using measurements from the

WindCube profiling lidar at the hub height of 80 m.

The wind direction histogram shown in Fig. 7a reveals a relatively uniform distribution of wind directions across the sector of interest for both the baseline and the controlled periods. Compared to the long-term wind speed distribution illustrated in Fig. 2, the wind speed histogram provided in Fig. 7b indicates above-average wind speeds during the experiment period; however, higher wind speeds are expected because most of the data were collected during the winter (in February and early

March, as illustrated in Fig. 4), when the wind resource is strong at the site. Figure 7c shows that similar turbulence intensity distributions were sampled during the baseline and the controlled periods. Last, the joint distribution of wind direction and wind speed during the experiment period is shown in Fig. 8, considering the total amount of data analyzed for the baseline and the controlled periods. For almost the entire wind direction sector, data were collected for wind speeds from 4–13 m/s.

## 3 Models

To help determine how accurately engineering wind farm control models represent wake steering in the field, the FLORIS tool and a probabilistic model of wind direction variability are used to predict realistic wake steering performance for the wind conditions observed during the experiment. In Section 3.1, we briefly describe the FLORIS model of the SMV wind plant, followed by a discussion in Section 3.2 of the model of wind direction variability.

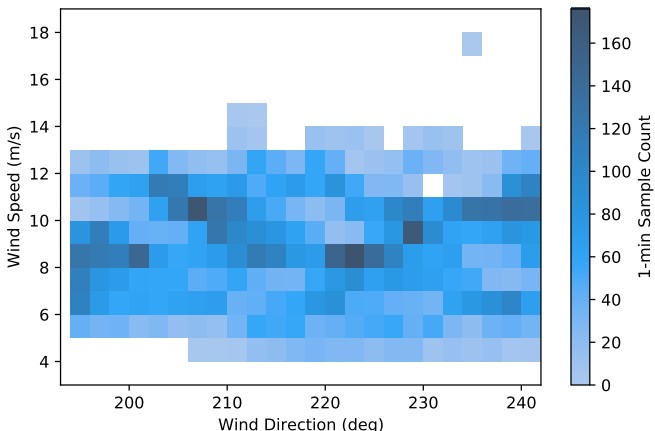

**Figure 8.** Number of 1-minute samples collected for the combined baseline and controlled periods grouped by wind direction and wind speed. Only bins containing samples from both the baseline and the controlled periods are shown.

## 3.1 FLORIS Wind Farm Control Engineering Model

Wake interactions are simulated for the SMV wind plant using the default Gauss-curl hybrid (GCH) model within FLORIS (King et al., 2021). The GCH model is built on the Gaussian wake deficit model presented by Bastankhah and Porté-Agel (2014) and Niayifar and Porté-Agel (2015), as well as the wake deflection model developed by Bastankhah and Porté-Agel (2016). To capture the effects that large-scale trailing vortices generated through yaw misalignment have on wake deflection, however, the GCH model includes a computationally efficient approximation of the curl model developed by Martínez-Tossas et al. (2019).

Although the curl-specific elements of the GCH model impact wake steering performance most significantly when wake steering is implemented along a row of several wind turbines, the most relevant feature for the two-turbine scenario investigated here is "yaw-added recovery," in which the vortices created by a misaligned wind turbine increase wake recovery in addition to deflecting the turbine's wake.

The FLORIS model of the SMV wind plant relies on the turbines' theoretical power and thrust coefficients, shown in Fig. 3,
to determine wake behavior, and it is further tuned using field measurements. As illustrated in Fig. 9, the theoretical power curve used in FLORIS closely matches the observed power curve for SMV6 during baseline operation using wind speed measurements from the WindCube Nacelle lidar. The FLORIS model is tuned to match the depth of the measured baseline wake losses for SMV5 during the experiment by adjusting the turbulence intensity input, which affects the rates of wake recovery and expansion; we found that when using the "gauss" velocity model, a turbulence intensity of 11% represents the
overall wake losses during the experiment reasonably well. Last, the power loss suffered as a result of yaw misalignment is modeled in FLORIS by scaling the rotor-averaged wind speed, $v_{\mathrm{avg}}$, used to determine power and thrust as follows:

$$v'_{\mathrm{avg}} = v_{\mathrm{avg}} \cos{(\gamma)}^{p_v/3} \qquad (1)$$



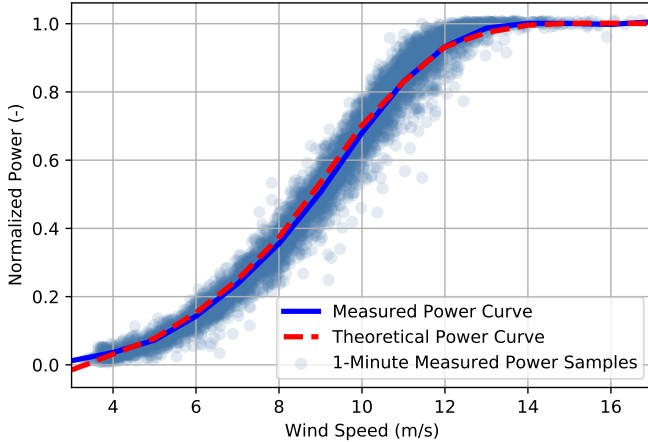

**Figure 9.** Measured and theoretical power curves for SMV6 during baseline operation normalized by rated power. Wind speed measurements are obtained from the WindCube Nacelle lidar at a range of 150 m. The data are filtered using the approach outlined in Section 4.1.

where $\gamma$ is the yaw misalignment, following the approach suggested by Bossanyi (2019). We use a cosine exponent of $p_v = 1.6$, estimated using WindCube ground-based lidar measurements and power data for SMV6 from previous experiments at the wind plant (Duc et al., 2017).

Examples of the hub-height flow fields generated by FLORIS for the SMV wind plant are provided in Fig. 10 for a wind speed of 8 m/s. Figure 10a shows the flow field with a wind direction of 195°, which is the southernmost wind direction investigated in this study. Figure 10b highlights the impact of SMV6 operating with a 20° yaw misalignment for a wind direction of 208°, which is the first wind direction (moving clockwise) for which yaw offsets are implemented (see Fig. 6). Finally, Fig. 10c shows the FLORIS flow field corresponding to the northernmost wind direction of 240° investigated here.

### 3.2 Wind Direction Variability

FLORIS is designed to model wake interactions for fixed wind directions and yaw positions. But in realistic dynamic wind environments with imperfect wake steering control, uncertainty exists in the yaw position a turbine achieves for a particular wind direction. Further, after the turbine settles on a specific yaw position, the wind direction will vary until the yaw error is large enough for the turbine to yaw again, causing wind direction uncertainty. Quick et al. (2017) and Quick et al. (2020) investigated the impact of yaw position uncertainty on optimal wake steering performance by performing FLORIS simulations with a distribution of possible yaw positions for a given wind direction. Similarly, Rott et al. (2018) and Quick et al. (2020) included wind direction uncertainty when optimizing wake steering control using FLORIS by assuming a distribution of possible wind directions about the intended wind direction. Here, we model the uncertainty resulting from wind direction variability and controller limitations using the approach presented by Simley et al. (2020b), wherein FLORIS simulations are performed assuming uncertainty in both yaw position and wind direction.



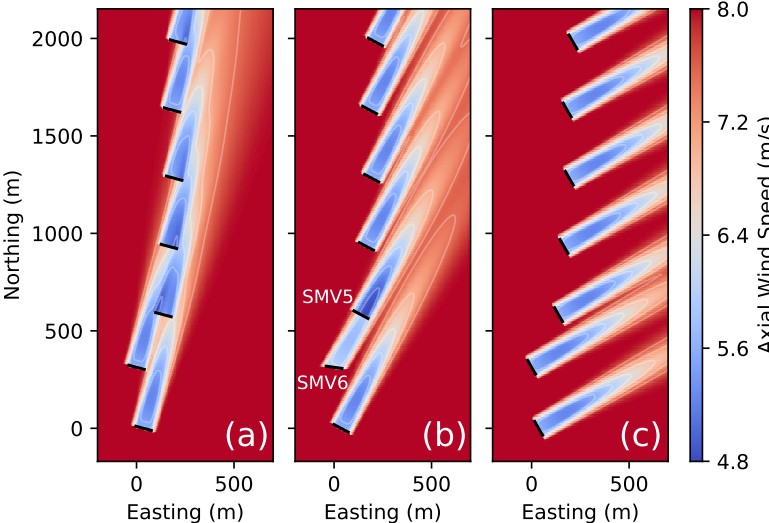

**Figure 10.** FLORIS flow fields for the SMV wind plant with a wind speed of 8 m/s; wind directions of (a) $195°$, (b) $208°$, and (c) $240°$; and corresponding target yaw offsets for SMV6 of (a) $0°$, (b) $20°$, and (c) $0°$.

As described in depth by Simley et al. (2020b) and Simley et al. (2020a), we model the impact of wind direction variability on wake steering performance by creating a joint probability mass function (PMF) of the wind directions and yaw positions for the controlled turbine (discretized using $1° \times 1°$ bins). First, the ideal PMF is established by assigning a probability of one to the intended yaw position corresponding to each wind direction using the yaw offset schedule. Next, the ideal PMF is convolved with a zero-mean joint PMF representing the uncertainty in the wind direction and yaw position (approximated as a bivariate normal distribution). Predicted mean yaw offsets can then be calculated from the resulting PMF by finding the expected value of the yaw position for a particular wind direction. Similarly, the predicted mean power production can be estimated by calculating the expected power from FLORIS across all possible yaw positions corresponding to the wind direction of interest.

The distribution of wind direction and yaw position uncertainty is characterized using the standard deviations of the wind direction uncertainty, $\sigma_\phi$, and the yaw position uncertainty, $\sigma_\theta$. As explained by Simley et al. (2020b), we assume a value of $\sigma_\theta = 1.75°$, which the authors found to closely approximate the yaw position uncertainty observed in simulations using a standard yaw controller. Assuming independent wind direction and yaw position uncertainty variables, $\sigma_\phi$ can be estimated from the standard deviation of the measured yaw misalignment, $\sigma_\gamma$, as:

$$\sigma_\phi = \sqrt{\sigma_\gamma^2 - \sigma_\theta^2}. \tag{2}$$

We explore different methods for estimating the yaw error standard deviation for SMV6 during baseline operation as a function of wind speed, as shown in Fig. 11. Specifically, we compare yaw error measurements using the turbine's nacelle wind vane, the WindCube Nacelle lidar, and the difference between the reference wind direction—defined as the average wind



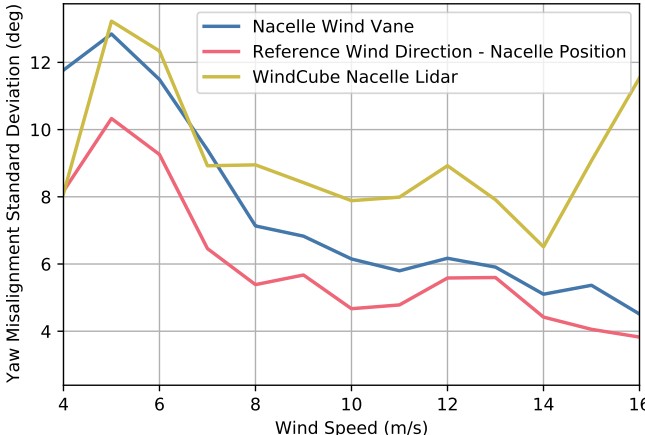

**Figure 11.** Standard deviation of 1-minute yaw misalignment measurements for SMV6 during baseline operation as a function of wind speed. Yaw misalignment is measured using the turbine's nacelle wind vane, the difference between the reference wind direction and the turbine's nacelle position, and the WindCube Nacelle lidar.

direction measured by turbines SMV1, SMV2, SMV3, and SMV7, as will be discussed in Section 4—and the turbine's nacelle position. Whereas the wind vane and nacelle lidar sample the more variable wind directions local to SMV6, the reference wind direction approximates the slowly varying mean wind direction across the wind plant; therefore, the latter method yields the lowest yaw misalignment standard deviation. Because the wind direction input to FLORIS is intended to represent the mean wind direction across the wind plant, we use the yaw error standard deviation determined from the reference wind direction as the input to the wind direction variability model. As shown in Fig. 11, the yaw misalignment standard deviation increases for wind speeds below 8 m/s because a) the wind direction tends to be more variable at lower wind speeds, and b) this turbine model's yaw controller is less responsive to wind direction changes for wind speeds below 7 m/s; therefore, we use wind speed-dependent yaw misalignment standard deviation values in the wind direction variability model, but we approximate the standard deviation as $\sigma_\gamma = 5.2°$ for all wind speeds greater than 8 m/s.

## 4 Data Processing

Before assessing the performance of wake steering, the measured 1-minute data are filtered to remove periods with abnormal wind turbine operation or poor data quality, which will be discussed in Section 4.1. Next, the reference wind direction, wind speed, and power variables are derived using measurements from turbines SMV1, SMV2, SMV3, and SMV7, as will be explained in Section 4.2. Last, Section 4.3 describes the procedure used for quantifying uncertainty in the wake steering performance metrics presented in Sections 5–7.





## 4.1 Filtering

To improve the likelihood that observed differences in power production for the baseline and the controlled periods are caused by wake steering rather than abnormal turbine operation, the data are filtered using the following steps. First, samples for which any of the test turbines (SMV5 and SMV6) or reference turbines (SMV1, SMV2, SMV3, and SMV7) are generating
less than 10 kW are removed. Although this step could remove baseline data supporting the argument that wake steering can prevent turbine shutdowns, as explained by Howland et al. (2019), it helps ensure that none of the turbines are unavailable or curtailed. Next, periods with known derating are removed from the data set. Additional periods with anomalous power production for any of the test or reference turbines are removed using the power curve filtering functions available in NREL's OpenOA software (Perr-Sauer et al., 2021). Finally, data within the first 10 minutes after switching between the baseline and
the controlled periods are removed to account for yaw controller transients.

For analyses relying on the WindCube Nacelle lidar measurements, the data are additionally filtered to remove 1-minute samples in which the availability of the underlying 4-Hz lidar measurements for any of the four beams is less than 35%. Note that because of periodic blade blockage, a significantly higher availability threshold would result in too much data removal.

## 4.2 Reference Variables

The reference wind direction, wind speed, and power variables are derived from turbines SMV1, SMV2, SMV3, and SMV7. The reference wind directions and wind speeds are used to represent the wind conditions encountered by the test turbines, whereas the reference power acts as an unbiased reference to which the power produced by the test turbines can be compared. Measurements from SMV6 are not used to estimate the wind direction or the wind speed to avoid the potentially confounding effects of yaw misalignment on the estimated values. SMV4 is excluded from the set of reference wind turbines because of
potential influences from wake steering, given its close proximity to the test turbines (see Fig. 1). Note that the WindCube ground-based lidar is not used to provide reference measurements either because of the impact of wakes on measurement accuracy for southerly flow.

The reference wind direction is calculated as the mean wind direction measured by SMV1, SMV2, SMV3, and SMV7 using their nacelle position sensors and wind vanes. Measurements from multiple turbines are averaged to smooth small-scale wind
direction variations that might be encountered at particular locations. The reference wind direction is calibrated to true north by first identifying the wind direction where the ratio between the mean power produced by SMV5 and SMV6 reaches a minimum, representing the direction where the wake losses suffered by SMV5 reach their peak. The offset between this wind direction and the known direction of alignment between SMV6 and SMV5 is then subtracted from the reference wind direction.

Similarly, the reference wind speed is based on the mean wind speed measured by SMV1, SMV2, SMV3, and SMV7 using
nacelle anemometry. We then apply additional steps to estimate the freestream equivalent wind speed encountered by the test turbines. First, to account for sensor bias, wake effects, and the impact of terrain and surface roughness on local wind conditions (e.g., the forest south of SMV7), wind direction and wind speed-dependent transfer functions are applied to the reference wind speeds to remove any bias from the wind speeds measured by SMV6 during baseline operation. This transfer





function is estimated as the ratio between the mean wind speed measured by SMV6 and the mean uncorrected reference wind
speed for the baseline periods, binned by wind direction (in overlapping 3° bins) and wind speed (in 1-m/s bins). Next, a nacelle
transfer function is applied to estimate the freestream wind speed from the nacelle anemometer-based reference wind speed.
The nacelle transfer function is calculated as the ratio between the mean wind speed measured by the WindCube Nacelle lidar
at a range of 150 m ($1.8D$) upstream of the rotor and the mean reference wind speed—considering only periods with baseline
control and wind directions with freestream inflow—in 1-m/s bins.

Last, the reference power is formed by averaging the power production of SMV1, SMV2, SMV3, and SMV7. Following an
approach similar to the reference wind speed derivation, a transfer function is applied to the average power produced by the
four reference wind turbines to remove any bias from the power generated by SMV6 during baseline operation (e.g., caused
by differences in turbine performance, wake effects, or the impact of local terrain and surface roughness). Again, this transfer
function is estimated by dividing the data into overlapping 3° wind direction bins as well as 1-m/s wind speed bins, then
calculating the ratio between the mean power produced by SMV6 and the mean uncorrected reference power for periods with
baseline control.

## 4.3   Uncertainty Quantification

To quantify uncertainty in the wake steering metrics presented in Section 5 through Section 7, we provide 95% confidence
intervals to accompany the estimates. Because many of the metrics require complicated calculations, analytic expressions for
the confidence intervals are difficult to derive; therefore, we use bootstrapping, wherein the collection of 1-minute data samples
used to derive a particular metric is randomly resampled with replacement many times to obtain a distribution of the estimates of
the metric (Dekking et al., 2005). From this distribution, which we derive using at least 2000 bootstrap samples, the confidence
interval containing 95% of the estimates is used as a measure of uncertainty. Many of the results presented in Section 5 through
Section 7 are derived for individual wind direction bins; for these cases, bootstrapping is performed using data from each wind
direction bin individually. Similarly, for metrics based on data from both the baseline and the wake steering periods, the data
corresponding to each control period are resampled independently before the final metric is calculated.

## 5   Yaw Offset Performance

In this section, we compare the yaw offsets achieved by the wake steering controller—measured using the nacelle wind vane
as well as the WindCube Nacelle lidar—to the ideal yaw offsets determined from the yaw offset schedule and the predicted
offsets based on the wind direction variability model. The overall mean achieved yaw offsets as a function of the reference
wind direction are presented in Section 5.1, whereas Section 5.2 highlights the achieved offsets for different wind speed bins.
Next, in Section 5.3, we directly compare the yaw offsets measured by the wind vane to those measured by the nacelle lidar.
Based on this comparison, we suggest corrections to the nacelle wind vane measurements.



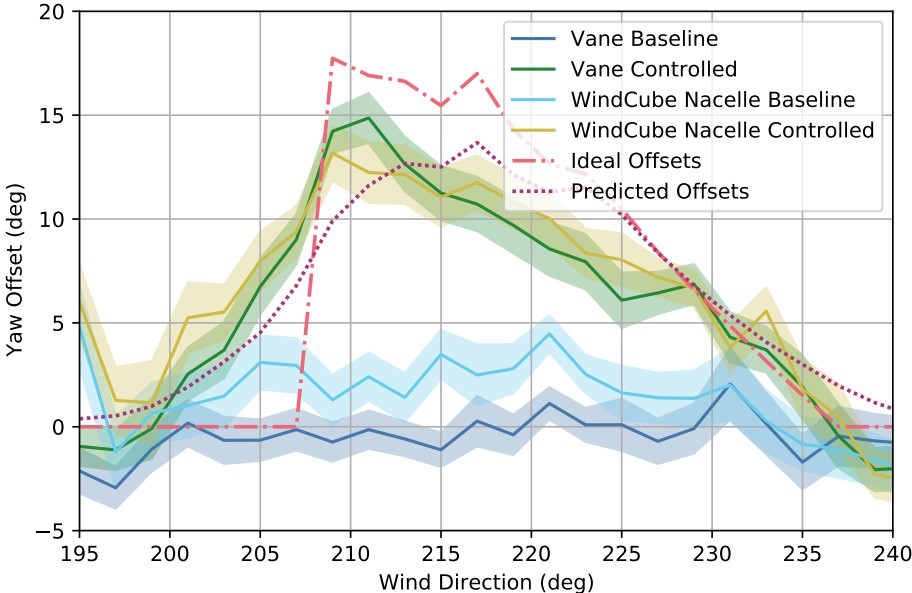

**Figure 12.** Mean yaw offsets for the baseline and the controlled operation for upstream turbine SMV6. Yaw offsets measured using the nacelle wind vane and WindCube Nacelle lidar are compared to the ideal (intended) yaw offsets and the predicted yaw offsets using the wind direction variability model. Shaded regions indicate the 95% confidence interval of the mean.

## 5.1 Achieved Yaw Offsets

The ideal, predicted, and achieved mean yaw offsets for all baseline and wake steering control periods are shown in Fig. 12 as a function of wind direction. As expected, the mean yaw offsets measured by the wind vane with baseline control are close to zero. But the mean offsets measured by the nacelle lidar show a bias of 2°–3° during baseline periods, suggesting that the wind vane might be poorly calibrated (as will be explored in more detail in Section 5.3). During wake steering control periods, similar mean yaw offsets are measured by the wind vane and nacelle lidar. Whereas the achieved yaw offsets fall short of the intended offsets for wind directions between 209° and 229°, they are reasonably well represented by the predicted yaw offsets using the wind direction variability model. Specifically, the wind direction variability model predicts a reduction in the peak yaw offsets accompanied by the unintended yaw offsets outside of the target wake steering sector. Although the achieved yaw offsets match this predicted trend, they exhibit a higher, more pronounced peak near the wind direction of 210°. Differences between the predicted and achieved yaw offsets could be partly explained by biases in the wind vane measurements (see Section 5.3), which propagate to the estimated wind direction signal used by the wake steering controller.





## 5.2 Wind Speed Dependence of Achieved Yaw Offsets

Whereas Fig. 12 revealed the mean yaw offsets aggregated among all wind conditions, Fig. 13 highlights the wind speed dependence of the ideal, predicted, and achieved yaw offsets. The ideal yaw offsets reflect the yaw offset schedule shown in Fig. 6; for wind speeds above 10 m/s, the target yaw offsets are gradually phased out, until no offsets are applied when the wind speed

reaches 14 m/s. The more spread out predicted yaw offsets show a similar reduction for wind speeds above 10 m/s, but they also vary for lower wind speeds because of the impact of the wind speed-dependent yaw error on the wind direction variability model, as explained in Section 3.2. Namely, because the standard deviation of the wind direction uncertainty increases for low wind speeds, the predicted yaw offsets reach a lower peak offset and are spread out over a wider wind direction sector as wind speed decreases below 8 m/s.

Despite the large scatter and uncertainty in the achieved yaw offsets for certain wind speed bins—caused by the relative lack of data for wind speeds below 8 m/s and above 12 m/s (see Fig. 7) as well as the greater wind direction variability for wind speeds below 8 m/s, potentially causing the reference wind direction measurements to poorly represent the wind conditions at SMV6—Fig. 13 reveals several trends. First, the positive wind vane bias observed during the baseline control, as measured by the nacelle lidar, is apparent for wind speeds below 10 m/s, but it disappears for higher wind speeds, suggesting wind speed-

dependent vane error. Additionally, when wake steering control is active, the achieved yaw offsets measured by the wind vane and nacelle lidar closely agree for the 8–10-m/s wind speed bin. But relative to the lidar-measured yaw misalignment, the wind vane appears to underestimate the yaw offsets at lower wind speeds and to overestimate the offsets at higher wind speeds.

Focusing on the wind speed dependence of the achieved and predicted yaw offsets with wake steering control, Fig. 13 shows that the achieved offsets for wind speeds below 8 m/s generally agree with the predicted offsets; because of the high wind

direction variability at these wind speeds, the achieved offsets are spread among a large wind direction sector. Uncertainty in the reference wind direction measurements stemming from wind direction variability could further contribute to the broadening of the achieved offset curves. For wind speeds between 8–10 m/s, the achieved yaw offsets closely match the predicted offsets. This wind speed bin is also favorable from a measurement perspective because of the large amount of data collected and the relatively low wind direction variability. Within the 10–12-m/s wind speed bin, the achieved yaw offsets tend to be higher than

predicted, possibly because of lower wind direction variability or larger yaw offsets persisting from operation in lower wind speeds. Last, for wind speeds between 12–14 m/s, measurement uncertainty caused by the relative lack of data obscures the yaw offset trends. But, as expected, the achieved offsets are relatively low (the maximum target yaw offset for this wind speed bin is only 5°).

## 5.3 Lidar-Based Validation of Yaw Offsets

Given the biases between the yaw offsets measured by the wind vane and the WindCube Nacelle lidar observed in Figs. 12 and 13, in this section we estimate wind speed-dependent transfer functions to correct the vane measurements. These transfer functions are determined by treating the yaw misalignment measured by the lidar as unbiased for all yaw offsets, although we assume that zero-mean lidar measurement errors exist (e.g., from the limitations of the wind field reconstruction based

**Figure 13.** Mean yaw offsets for the baseline and the controlled operation for upstream turbine SMV6 binned by wind speed. Yaw offsets measured using the nacelle wind vane and WindCube Nacelle lidar are compared to the ideal (intended) yaw offsets and the predicted yaw offsets using the wind direction variability model. Shaded regions indicate the 95% confidence interval of the mean.

on line-of-sight wind speed measurements). To increase the amount of data that can be analyzed, the transfer functions are

determined using reference wind directions up to 270°. But to reduce the chance that wake effects from SMV7 influence the yaw misalignments measured by the vane or lidar (see Fig. 1), only data corresponding to wind directions above 210° are included in the analysis.




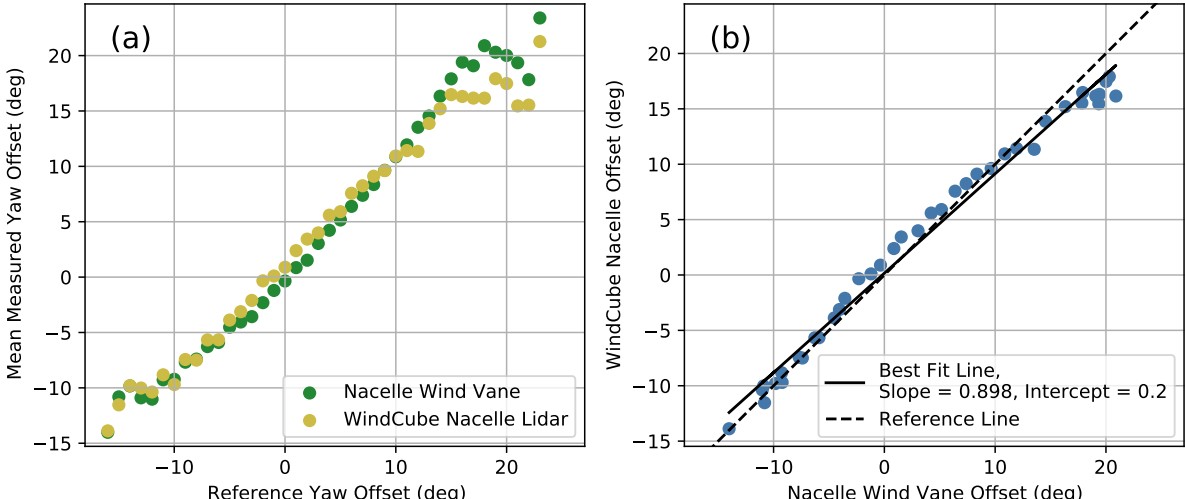

**Figure 14.** (a) Mean yaw offsets for SMV6 measured by nacelle wind vane and WindCube Nacelle lidar binned by reference yaw offset (the difference between the reference wind direction and the nacelle position of SMV6). (b) Relationship between the yaw offsets measured by the nacelle wind vane and the WindCube Nacelle lidar along with the best-fit line.

Corrective wind vane transfer functions are estimated using least-squares regression by fitting a line to the lidar-measured yaw offsets as a function of the wind vane-measured offsets. The best-fit slope and intercept could then be applied to the wind vane measurements to remove measurement biases; however, applying linear regression to the individual 1-minute yaw offset measurements poses a problem because we assume that both the lidar-measured yaw misalignments and the offsets measured by the wind vane contain random measurement errors. When the predictor variable in a linear regression contains measurement errors, the regression slope tends to be underestimated through a process called "regression dilution" (Frost and Thompson, 2000). To overcome this challenge, we attempt to remove zero-mean measurement errors by first binning and averaging the yaw offset measurements. Specifically, we bin the measured yaw offsets by a separate reference yaw offset, given by the difference between the reference wind direction and the nacelle position of SMV6 (see Fig. 14a). With the random measurement errors significantly reduced, the bin-averaged yaw offsets measured by the lidar and wind vane can then be compared to reveal biases in the wind vane measurements. An example of a linear regression applied to the bin-averaged yaw offsets for all wind speeds is provided in Fig. 14b, revealing an intercept of only $0.2°$ but a slope of 0.9, indicating the tendency for the wind vane to overestimate the magnitude of the yaw misalignment.

Scatter plots of the bin-averaged yaw offsets measured by the WindCube Nacelle lidar and the wind vane, along with the corresponding best-fit lines, are provided in Fig. 15 for different wind speed bins. Beginning with the wind vane bias when the measured yaw misalignment is zero, indicated by the intercept, a positive bias of $\sim 2°$ is observed for wind speeds below 8 m/s. For the 8–10-m/s wind speed bin, no significant bias is observed; however, for wind speeds above 10 m/s, the wind vane measurement contains a negative bias of approximately $-3°$. This wind speed dependence of the mean wind vane measurement



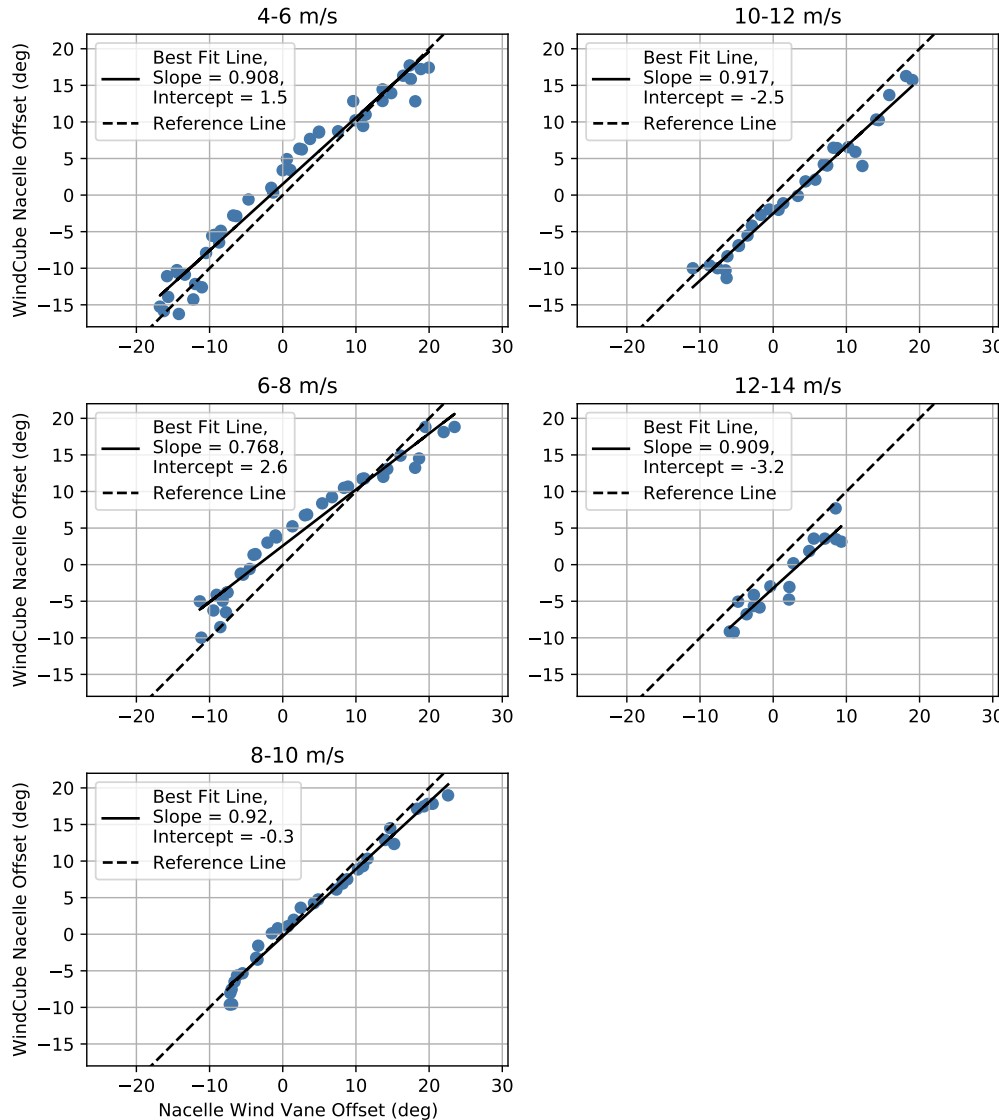

**Figure 15.** Relationship between yaw offsets for SMV6 measured by the nacelle wind vane and the WindCube Nacelle lidar binned by wind speed along with best-fit lines.

error has been previously reported by Kragh and Hansen (2015). Despite the measurement bias for individual wind speed bins, it is likely that the wind vane measurements are calibrated to achieve an average measurement bias close to zero, as indicated by the results in Fig. 14. In contrast to the intercepts, the slopes of the best-fit lines in Fig. 15 are nearly constant across different wind speeds; aside from an outlier in the 6–8-m/s wind speed bin, a slope of $\sim 0.9$ is observed, suggesting that the wind vane measurements overestimate the true yaw misalignment by roughly 10%.





Although we did not apply any corrections to the wind vane signals used by the wake steering and yaw controllers in this experiment, the identified transfer functions could be used to help ensure that the intended yaw offsets are achieved in future controller implementations. Note that the amount of wind vane bias likely depends on the wind turbine model.

## 6 Impact of Yaw Misalignment on Power Production

Understanding the relationship between yaw misalignment and power production is paramount to the design of optimal wake steering strategies. In this section, we use measurements from the WindCube Nacelle lidar to estimate the impact of yaw misalignment on power production. Whereas FLORIS models the power loss from yaw misalignment via the effective wind speed (see Eq. 1), in this section we use the more traditional method of identifying the cosine exponent, $p_P$, used to model the ratio between the power with yaw misalignment, $P$, and the power that would have been produced during aligned operation,
$P_0$:

$$P/P_0 = \cos(\gamma)^{p_P}. \tag{3}$$

A wide range of $p_P$ estimates are presented in the literature. For example, Fleming et al. (2017) estimate $p_P = 1.43$ using data from a commercial wind plant, Gebraad et al. (2016) find that a value of $p_P = 1.88$ fits results from LES simulations, and Medici (2005) determines a value of $p_P = 2$ from a wind tunnel experiment. But as discussed by Liew et al. (2020),
blade element momentum theory predicts $p_P = 3$. Note that $p_P$ and the value of $p_v$ used in FLORIS (see Eq. 1) are not necessarily equivalent, but they are expected to agree in below-rated wind speeds when the wind turbine's coefficient of power is maximized.

To estimate $p_P$ using Eq. 3, we measure the yaw misalignment, $\gamma$, using the nacelle lidar. Rather than treating the reference power defined in Section 4.2 as $P_0$, we use the power given by the theoretical power curve shown in Fig. 3 combined with
the lidar-measured wind speed. Using this nacelle lidar-based reference power helps ensure that the $P_0$ estimates represent the wind conditions local to SMV6. The exponent $p_P$ is then estimated by fitting the function $\cos(\gamma - \alpha)^{p_P}$ to the ratio between the mean power produced by SMV6 and the mean value of $P_0$ binned by $\gamma$ using nonlinear least-squares optimization. Note that $\alpha$ is treated as an independent variable in the curve-fitting procedure and represents the yaw misalignment where power is maximized. As explained in Section 5.3, data corresponding to reference wind directions from 210°–270° are used in this
analysis.

The ratios between the mean power produced by SMV6 and the mean value of the lidar-estimated $P_0$, along with the best-fit $\cos(\gamma - \alpha)^{p_P}$ curves, are shown in Fig. 16 for different wind speed bins. The highest cosine exponent of $p_P \approx 2.5$ is estimated for wind speeds from 4–8 m/s. Region 2 of the wind turbine's power curve, in which the controller tracks the optimal tip-speed ratio to maximize power production, roughly spans wind speeds from 5–8 m/s; therefore, the maximum value of $p_P$ is expected
in this wind speed range. As wind speed increases above 8 m/s, the estimated value of $p_P$ decreases from ∼1.15–1.3 for wind speeds between 8–12-m/s to $p_P = 0.35$ for the 12–14-m/s wind speed bin. Once the wind turbine reaches rated power in the 14–16-m/s wind speed bin, the estimated $p_P$ is close to zero, with power showing almost no dependence on yaw misalignment





(albeit based on limited data). The reduction of the cosine exponent $p_P$ as the wind speed increases above Region 2 of the power curve agrees with the shape of the power curve, shown in Fig. 3. As the wind speed increases, the slope of the power curve decreases; thus, reductions in the effective rotor-averaged wind speed caused by yaw misalignment should result in a smaller change in power. Note that by modeling the relationship between the power and the yaw misalignment via the effective wind speed (see Eq. 1), FLORIS captures the *trend* of the observed wind speed dependence of $p_P$, if not necessarily the exact relationship. The large reduction in $p_P$ as wind speed increases above Region 2 could have significant implications for wake steering strategies; in some situations, wake steering might be most effective at higher wind speeds where less power is lost from yaw misalignment. Finally, despite the observed trends in the $p_P$ values, we note that the estimation uncertainty is high. For example, in the 6–8-m/s wind speed bin, the 95% confidence interval of the estimated cosine exponent of $p_P \approx 2.5$ ranges from roughly 2–3.

The estimated cosine exponent of $p_P \approx 2.5$ for wind speeds below 8 m/s is significantly greater than the value of $p_P \approx 1.6$ estimated prior to the experiment; however, the cosine exponent of $p_P \approx 1.6$ was estimated using 10-minute data samples for wind speeds between 5–9 m/s (Duc et al., 2017). Measurements for wind speeds above 8 m/s likely contributed to the lower cosine exponent (see Fig. 16). Further, because 10-minute power measurements corresponding to small yaw misalignments contain contributions from a wider range of instantaneous yaw offsets than 1-minute measurements, the average power production is expected to be less; therefore, the peak of the power curve as a function of yaw offset is likely flatter when using 10-minute data, leading to smaller best-fit cosine exponents.

Last, although Fig. 16 reveals the impact of yaw misalignment on power production for a specific wind turbine, recent research suggests that the relationship between yaw misalignment and power depends on the atmospheric boundary layer as well as the turbine's control system. Howland et al. (2020) show how the power production of a misaligned wind turbine depends on the wind shear and veer profiles interacting with the rotor, which can introduce asymmetry in the relationship between yaw misalignment and power. The authors also explain how the generator torque control logic used during below-rated operation can influence the impact of yaw misalignment on power via changes in rotor speed. Further, using CFD simulations, Cossu (2021) demonstrates that re-optimizing the blade pitch angle of a yawed wind turbine in below-rated operation can both reduce the power loss from yaw misalignment and increase the power gain from wake steering at downstream turbines. Through a wake steering field experiment, Doekemeijer et al. (2021) observe asymmetry in the power loss as a function of yaw misalignment, similar to the findings of Howland et al. (2020), while also noting a relatively flat peak of the power curve as a function of yaw offset followed by a sharp drop in power production for more extreme offsets.

## 7 Energy Improvement from Wake Steering

To assess the impact of wake steering on the performance of the test turbines SMV5 and SMV6, we measure the change in energy production between the baseline and the wake steering control periods as a function of wind direction. Specifically, after dividing the measurement data into 2°-wide wind direction bins, we quantify the ratio between the energy produced by the test turbines and the reference turbines for the baseline and the wake steering control periods using the *balanced energy*

**Figure 16.** Normalized power of SMV6 as a function of yaw offset measured using the WindCube Nacelle lidar binned by wind speed. Best-fit cosine power law curves are provided along with the best-fit cosine exponents and wind direction offsets, with accompanying 95% confidence intervals. Shaded regions indicate the 95% confidence intervals of the energy ratios for individual yaw offset bins (blue) and the best-fit cosine power law curves (red).

*ratio* method introduced by Fleming et al. (2019). For each wind direction bin, the energy ratio is calculated as:

$$R_{\mathrm{Energy}} = \frac{\sum_{i=1}^{N_{\mathrm{ws}}} w_i \overline{P}_{\mathrm{Test},i}}{\sum_{i=1}^{N_{\mathrm{ws}}} w_i \overline{P}_{\mathrm{Ref},i}} \qquad (4)$$



where $\overline{P}_{\mathrm{Test},i}$ and $\overline{P}_{\mathrm{Ref},i}$ are the mean test and reference powers, respectively, in wind speed bin $i$, and the weighting factor $w_i$ is defined as the total number of samples in wind speed bin $i$ for the baseline and the controlled periods combined. The test

power $P_{\mathrm{Test}}$ can be the power produced by the downstream turbine SMV5, the power of the upstream turbine SMV6, or the average power produced by SMV5 and SMV6. $P_{\mathrm{Ref}}$ is given by the reference power defined in Section 4.2. The weights $w_i$ ensure that the energy ratio is based on the observed distribution of the wind speeds while providing a fair comparison between the measurements from the baseline and the controlled periods. Normalization by the reference power is performed to attempt to control for factors beyond wake steering that could cause performance to change, such as wind shear or turbulence intensity,

assuming the test and reference turbines are affected equivalently.

In Section 7.1, we present the overall energy ratios for the baseline and the controlled periods as well as the change in the energy ratio with wake steering for the downstream turbine, SMV5, the upstream controlled turbine, SMV6, and the two turbines combined. In Section 7.2, the energy ratios and the changes in energy ratio with wake steering are shown for individual wind speed bins, highlighting wind speeds where wake steering is most effective. Finally, in Section 7.3, we estimate the long-

term change in energy production from wake steering for the combined turbines using the long-term wind rose for the site, shown in Fig. 2. For all scenarios, the measured energy ratios and changes in energy production are compared to estimates using the FLORIS model.

## 7.1 Energy Gain

The overall energy ratios and the change in energy ratio for the baseline and the wake steering control periods for the down-

stream turbine, SMV5, are plotted in Fig. 17 as a function of wind direction, along with 95% confidence intervals. The measured energy ratios and the change in energy ratio with wake steering are compared to the same metrics based on FLORIS simulations using three different FLORIS modeling assumptions. First, FLORIS estimates of power production are calculated for the observed distribution of wind directions, wind speeds, and yaw offsets measured using SMV6's nacelle wind vane (labeled "FLORIS" in Fig. 17). Next, the ideal FLORIS estimates are calculated using the intended yaw offsets for SMV6 as a function

of observed wind direction and wind speed according to the yaw offset schedule shown in Fig. 6. Last, the realistic predicted energy ratios based on FLORIS are calculated by combining the intended yaw offsets for SMV6 with the wind direction variability model discussed in Section 3.2.

As shown in Fig. 17, improvements in the energy production of SMV5 from wake steering are observed for wind directions from roughly 205°–225°, with a peak gain of nearly 0.15 (i.e., 15% of the average energy production of the unwaked reference

turbines) at 213°. The measured energy gains are generally greater than the FLORIS estimates based on the observed distribution of the yaw offsets as well as the predicted FLORIS gains, but, as expected, they are lower than the ideal FLORIS gains. A loss in energy is observed in the 203° wind direction bin, as predicted to a lesser extent by the wind direction variability model. Because of unintended yaw offsets for wind directions below 208° (see Fig. 12) the wake of SMV6 can potentially be redirected back toward SMV5, causing a loss in energy production (i.e., "wrong-way steering"). Note that although a similar

loss in power for wind directions immediately below the intended wake steering sector was observed in a previous experiment (Fleming et al., 2020; Simley et al., 2020a), the power loss shown in Fig. 17 coincides with a wind direction bin where





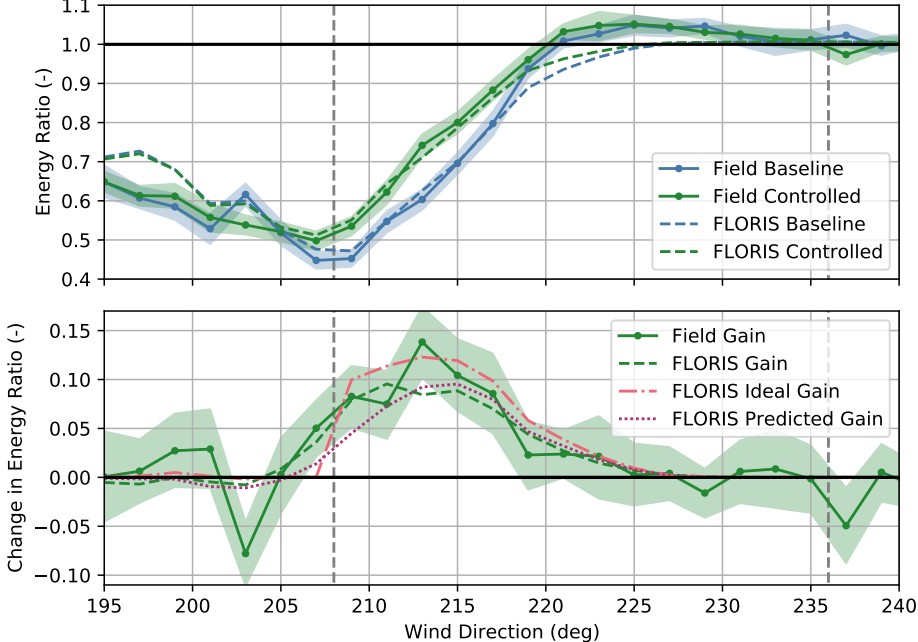

**Figure 17.** Energy ratios for the baseline and the controlled periods and the change in the energy ratio for the downstream turbine, SMV5. Energy ratios are derived from field observations as well as from FLORIS calculations using 1) observed yaw offsets, 2) ideal (intended) yaw offsets, and 3) predicted yaw offsets using the wind direction variability model. Shaded regions indicate the 95% confidence intervals of the energy ratios for individual wind direction bins. The gray dashed lines encompass the intended wake steering sector.

the baseline energy production appears anomalously high; therefore, the loss could potentially be an artifact of significantly different operating conditions experienced by the wind turbines during the baseline and the controlled periods. Finally, as evidenced by energy ratios greater than one for both the baseline and the controlled periods, a slight speedup effect at the edge of

the wake likely exists for wind directions between approximately 221°–233°.

The energy ratios for the upstream turbine, SMV6, shown in Fig. 18, reveal energy losses from the yaw misalignment of up to ∼5% when wake steering is active. Losses are observed for wind directions from 199°–237°, corresponding to the sector where nonzero-mean yaw offsets are observed, as shown in Fig. 12. The greatest energy losses occur between 211°–217°, near where the highest achieved yaw offsets occur. The measured reductions in energy production are roughly in line with the

predicted FLORIS losses using the wind direction variability model, but they are generally lower than the ideal FLORIS losses based on the intended yaw offsets; however, significant measurement uncertainty resulting from the relatively small changes in energy production precludes a direct comparison between the observed and the predicted energy losses as a function of wind direction.

To reveal the net impact of wake steering on energy production as a function of wind direction, the energy ratios for the

baseline and the controlled periods along with the change in energy ratio for the average power produced by SMV5 and SMV6





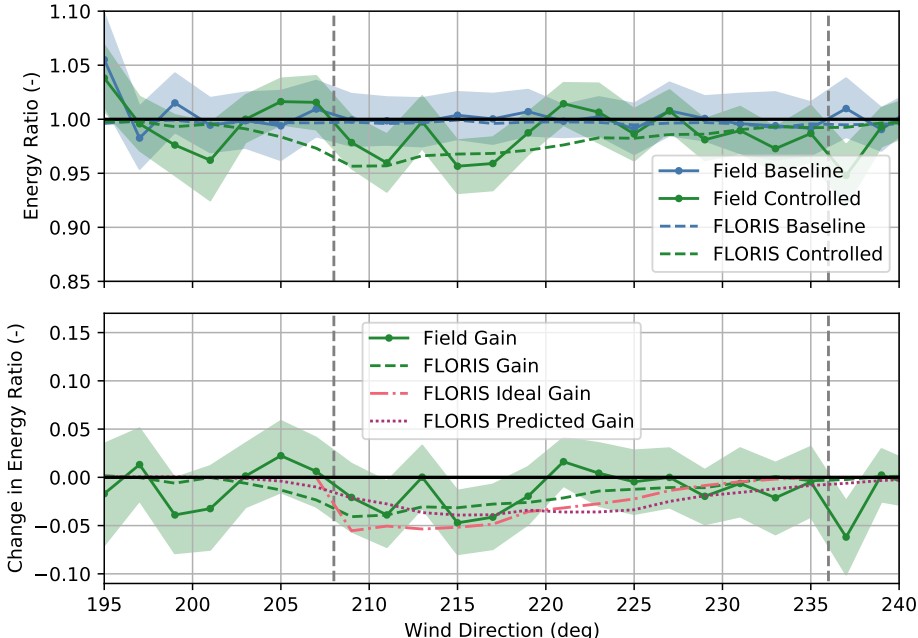

**Figure 18.** Energy ratios for the baseline and the controlled periods and the change in the energy ratio for the upstream turbine, SMV6. Energy ratios are derived from field observations as well as from FLORIS calculations using 1) observed yaw offsets, 2) ideal (intended) yaw offsets, and 3) predicted yaw offsets using the wind direction variability model. Shaded regions indicate the 95% confidence intervals of the energy ratios for individual wind direction bins. The gray dashed lines encompass the intended wake steering sector.

are provided in Fig. 19. Despite the energy loss at SMV6 from the yaw misalignment, a net increase in energy production of up to 3%–5% of the energy produced by the unwaked reference turbines is measured for wind directions from 205°–225°. The increases in energy production in this sector are generally greater than both the predicted and the ideal FLORIS gains. As predicted using FLORIS combined with the wind direction variability model, losses are observed near 207°—possibly because of unintentional wrong-way steering—and above 225°. In the latter case, a minor loss in energy is predicted by FLORIS because the gains at SMV5 are not large enough to outweigh the loss in energy from the yaw misalignment at SMV6.

Because increases in energy production are observed for the combined upstream and downstream turbines for most wind directions, as shown in Fig. 19, but losses are measured as well, we assess the net impact of wake steering on energy production over the entire wind direction sector from 195°–240°. We quantify the net impact by estimating the percentage of wake losses in the sector that are reduced by wake steering, as discussed by Fleming et al. (2020). Wake losses are calculated for the baseline and the controlled periods separately by first binning the difference between the reference power and the average power produced by SMV5 and SMV6 by wind direction and wind speed. Next, the power losses are weighted by the fraction of the total samples contained in each bin, including the baseline and the controlled periods. The weighted power losses are then summed to determine the average wake losses over the entire wind direction sector. Using this method, wake steering



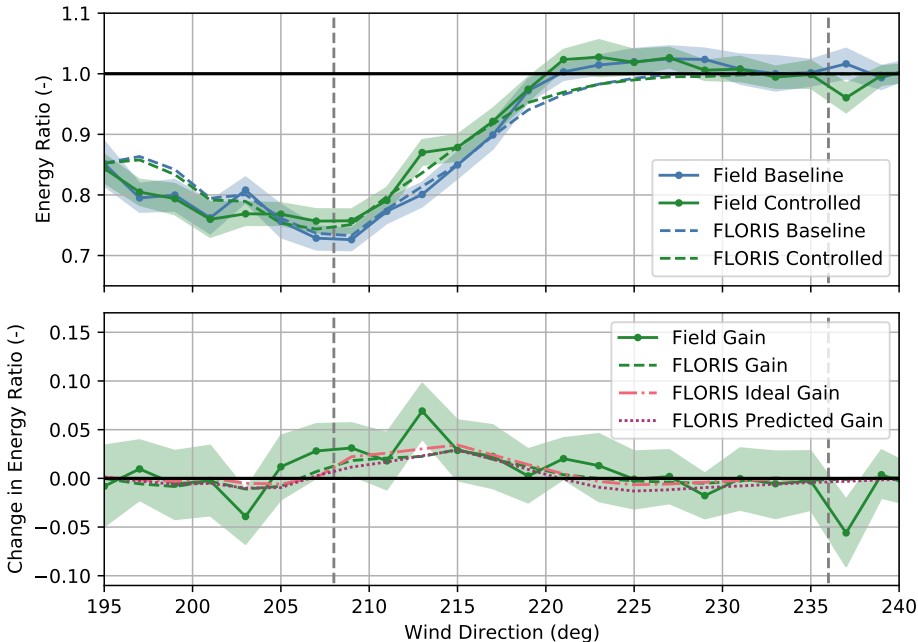

**Figure 19.** Energy ratios for the baseline and the controlled periods and the change in the energy ratio for turbines SMV5 and SMV6 combined. Energy ratios are derived from field observations as well as from FLORIS calculations using 1) observed yaw offsets, 2) ideal (intended) yaw offsets, and 3) predicted yaw offsets using the wind direction variability model. Shaded regions indicate the 95% confidence intervals of the energy ratios for individual wind direction bins. The gray dashed lines encompass the intended wake steering sector.

was found to reduce wake losses by 5.7% but with a large 95% confidence interval of -0.3% to 11.5%, estimated through bootstrapping.

## 7.2  Wind Speed Dependence of Energy Gain

The changes in the energy ratios with wake steering for wind turbines SMV5 and SMV6 combined for different wind speed bins are provided in Fig. 20, followed by the wind speed-dependent energy ratio changes for the downstream turbine, SMV5,

and upstream turbine, SMV6, separately, in Figs. 21 and 22, respectively. For reference, the energy ratios for the combined upstream and downstream wind turbines during the baseline and the controlled periods are provided in Fig. A1 in Appendix A. Note that because fewer data samples comprise the energy ratios for the individual wind speed bins, the uncertainty is larger than that of the overall energy ratios shown in Section 7.1. Nevertheless, trends in wake steering performance as a function of wind speed can be observed.

Mixed wake steering performance is observed for wind speeds below 8 m/s, as shown in Fig. 20. For the 4–6-m/s wind speed bin, significantly larger increases in energy production are achieved with wake steering than predicted by FLORIS, with energy ratios increasing by as much as 0.1–0.3 for wind directions from 195°–233°. The large energy gains for wind



**Figure 20.** Change in the energy ratios for the baseline and the controlled periods for turbines SMV5 and SMV6 combined binned by wind speed. Energy ratios are derived from field observations as well as from FLORIS calculations using 1) observed yaw offsets, 2) ideal (intended) yaw offsets, and 3) predicted yaw offsets using the wind direction variability model. Shaded regions indicate the 95% confidence intervals of the energy ratios for individual wind direction bins. The gray dashed lines encompass the intended wake steering sector.

speeds below 6 m/s are potentially caused by wake steering preventing SMV5 from shutting down by allowing higher velocity inflow to interact with the rotor, as discussed by Howland et al. (2019). As shown in Figs. 21 and 22, however, significant
energy improvements are observed for the upstream wind turbine, SMV6, in addition to SMV5 for the 4–6-m/s wind speed bin; therefore, additional sources of the apparent energy improvement for this wind speed bin might exist. For wind speeds





**Figure 21.** Change in the energy ratios for the baseline and the controlled periods for the downstream turbine, SMV5, binned by wind speed. Energy ratios are derived from field observations as well as from FLORIS calculations using 1) observed yaw offsets, 2) ideal (intended) yaw offsets, and 3) predicted yaw offsets using the wind direction variability model. Shaded regions indicate the 95% confidence intervals of the energy ratios for individual wind direction bins. The gray dashed lines encompass the intended wake steering sector.

between 6–8 m/s, wake steering appears to cause little net change in energy production for the combined wind turbines despite the large energy gains predicted using FLORIS. Specifically, as illustrated in Figs. 21 and 22, lower-than-predicted energy gains at SMV5 are roughly cancelled by losses at SMV6. Poor wake steering performance for wind speeds between 6–8 m/s is

likely related to relatively high wind direction variability, as explained in Section 3.2.




**Figure 22.** Change in energy ratios for the baseline and the controlled periods for the upstream turbine, SMV6, binned by wind speed. Energy ratios are derived from field observations as well as from FLORIS calculations using 1) observed yaw offsets, 2) ideal (intended) yaw offsets, and 3) predicted yaw offsets using the wind direction variability model. Shaded regions indicate the 95% confidence intervals of the energy ratios for individual wind direction bins. The gray dashed lines encompass the intended wake steering sector.

As revealed in Fig. 20, significant energy gains from wake steering are observed for wind speeds from 8–12 m/s. For the 8–10-m/s wind speed bin, the energy improvement is roughly in line with the energy gains predicted using FLORIS. Notably, large gains at SMV5, shown in Fig. 21, outweigh the relatively high losses at SMV6, illustrated in Fig. 22. Note that the individual changes in energy ratio from wake steering for SMV5 and SMV6 closely match the FLORIS-based predictions for this wind





speed bin as well. For the 10–12-m/s wind speed bin, the energy gains at SMV5 are slightly greater than the improvements predicted using FLORIS, yet very little change in energy is observed at SMV6 from yaw misalignment. Consequently, a greater-than-predicted net energy gain is observed for the wind turbine pair.

Last, the performance of wake steering for the 12–14-m/s wind speed bin is unclear based on the data collected. Figure 20 suggests that energy ratio increases of up to 0.1 are achieved for wind directions between 213°–221°. But these gains appear to

be largely cancelled by losses for wind directions from 201°–211°, which could be caused by wrong-way steering. On the other hand, the target yaw offsets are limited to a maximum of 5° for wind speeds between 12–14 m/s; consequently, the FLORIS predictions show very little change in energy production across the wind direction sector investigated. Further, because the yaw controller is less responsive to small yaw offset commands, it might not be able to effectively implement the intended offsets of 5° or less. Finally, the large variations in the observed changes in energy ratio for different wind directions could be caused

by the relative lack of data collected for wind speeds above 10 m/s, as indicated in Fig. 7b.

### 7.3 Long-Term Corrected Energy Gain

As discussed in Section 2.3, above-average wind speeds were observed during the experiment period because of the typically stronger wind resource at the site in the winter. To estimate the expected impact of wake steering on energy production during a typical year, we compute long-term corrected energy ratios for the baseline and the wake steering periods based on the long-

term wind rose frequencies shown in Fig. 2. The long-term corrected energy ratio calculation requires only a slight modification to the energy ratio definition in Eq. 4; instead of weighting the mean power for the test and reference turbines in a particular wind direction and wind speed bin by the total number of samples measured in that bin, we weight the mean power by the long-term frequency of occurrence of the wind conditions represented by the bin.

Using the modified energy ratio calculations, the long-term corrected energy ratios for the baseline and the wake steering

periods together with the change in energy ratio with wake steering for SMV5 and SMV6 combined are provided in Fig. 23. Energy gains of up to 3%–5% of freestream energy production are measured for wind directions between 207–225°, generally matching the predicted energy improvements using FLORIS. Slight losses in energy production are observed for wind directions below 207° and above 225°, as also predicted by FLORIS. Overall, the long-term corrected change in energy ratio from wake steering follows the same trends as the change in energy ratio for the experiment period shown in Fig. 19; however, the

peak gains and losses are less extreme when the long-term corrected wind rose is applied.

To estimate the long-term net impact of wake steering for the combined upstream and downstream wind turbines over the entire wind direction sector from 195°–240°, we repeat the analysis of the wake loss reduction from wake steering introduced in Section 7.1. But similar to the long-term corrected energy ratio method, we modify the procedure for calculating the wake losses by weighting the difference between the mean power of the reference and test turbines in each wind direction and wind

speed bin by the long-term frequencies of occurrence of the wind conditions rather than by the frequencies observed during the experiment period. Based on this procedure, we estimate a long-term corrected wake loss reduction of 9.8% from wake steering but with a large 95% confidence interval spanning from 2.2%–16.6%. The expected long-term corrected wake loss reduction from wake steering is greater than the value of 5.7% calculated for the experiment period in part because wind speeds



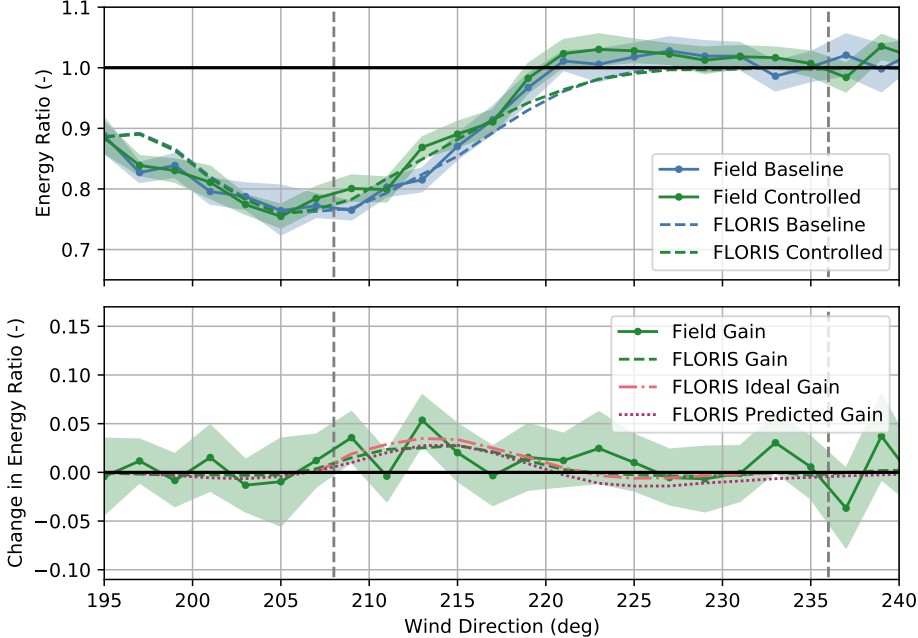

**Figure 23.** Long-term corrected energy ratios for the baseline and the controlled periods and the change in energy ratio for turbines SMV5 and SMV6 combined. Energy ratios are derived from field observations combined with the long-term wind rose shown in Fig. 2 as well as from FLORIS calculations using 1) observed yaw offsets, 2) ideal (intended) yaw offsets, and 3) predicted yaw offsets using the wind direction variability model. Shaded regions indicate the 95% confidence intervals of the energy ratios for individual wind direction bins. The gray dashed lines encompass the intended wake steering sector.

from 4–6 m/s occur more frequently in the long-term wind rose. Although the wind turbines' power production is relatively
low for this wind speed bin, the large energy improvements from wake steering contribute to a significantly higher reduction in overall wake losses.

## 8   Conclusions

In this paper, we analyzed the performance of wake steering control for two wind turbines spaced $3.7D$ apart at a commercial wind plant by examining the change in energy production from wake steering as well as the achieved yaw offsets during the
3-month experiment period. To highlight the wind speed dependence of wake steering performance, we presented results in aggregate as well as for individual wind speed bins between 4–14 m/s. The overall improvement in energy production was quantified by estimating the percentage of wake losses reduced by wake steering, both during the experiment period and extended to represent the long-term wind resource for the site. To help validate the use of the FLORIS engineering wind farm control tool for wake steering controller design, we compared the measured energy production to the FLORIS predictions
based on the achieved yaw offsets, the ideal offsets, and the predicted yaw offsets using a model of wind direction variability.





We also compared the achieved yaw offsets to the predicted offsets based on the wind direction variability model. Finally, we used measurements from a WindCube Nacelle lidar to determine the accuracy of the nacelle wind vane used to implement wake steering as well as to better understand the power loss caused by the yaw misalignment.

Overall energy gains of up to 3%–5% of the potential freestream energy production were observed for the combined upstream and downstream wind turbines for specific wind directions, resulting in an estimated 5.7% net reduction in wake losses over the wind direction sector investigated. Large energy improvements from wake steering were observed for wind speeds from 4–6 m/s, likely because the higher inflow velocities at the downstream wind turbine prevented it from shutting down as frequently; however, poor wake steering performance was measured for wind speeds from 6–8 m/s as a result of significant wind direction variability and power loss from the yaw misalignment. On the other hand, significant energy improvements were observed for wind speeds between 8–12 m/s. Because of smaller target yaw offsets and a relative lack of data, the effectiveness of wake steering in the 12–14-m/s wind speed bin—just below the turbines' rated wind speed—is inconclusive. After correcting the change in energy from wake steering during the experiment period using the long-term site wind rose, we estimated that the wake loss reduction from wake steering increases to 9.8% for the wind directions analyzed. Despite the large uncertainty in the estimated reductions in wake losses, this improvement underscores the importance of assessing wake steering performance in wind conditions representative of the long-term wind resource at the site.

As revealed by the analysis of nacelle lidar measurements, the wind speed dependence of wake steering performance largely stems from the impact of yaw misalignment on power production. For wind speeds from 4–8 m/s—encompassing Region 2 of the power curve, wherein power is maximized—the greatest power loss from yaw misalignment was measured. But for wind speeds from 8–12 m/s, the power loss for a given yaw misalignment is roughly half the loss observed in Region 2. As wind speed increases to 12–14 m/s—just below rated wind speed for the turbine investigated—almost no impact on power production from yaw misalignment was detected. Thus, aside from the benefits observed near the cut-in wind speed, the greatest opportunities for wake steering might be at higher wind speeds where a significantly lower penalty is incurred when operating misaligned with the wind.

Further, yaw misalignment measurements from the nacelle lidar revealed a wind speed-dependent bias in the wind vane measurements as well as the tendency for the wind vane to overestimate the true yaw misalignment. Although we found that the resulting wind vane measurement error was typically within a few degrees, we suggest that sensors used to measure yaw misalignment as part of a wake steering control strategy be carefully calibrated to maximize the effectiveness of wake steering.

When combined with the wind direction variability model, we found that FLORIS predicts the impact of wake steering on energy production reasonably well. But for the combined upstream and downstream wind turbines, the observed energy gains were generally higher than those predicted by FLORIS. Note that we used a relatively high turbulence intensity of 11% in the FLORIS model to match the measured baseline wake losses. The energy gain predictions from FLORIS could be improved by using a lower turbulence intensity value (the effectiveness of wake steering improves as turbulence intensity decreases), suggesting that further work is needed to reconcile the wake deficit and wake deflection models in FLORIS, at least for relatively short turbine separations such as the $3.7D$ spacing investigated here. The wind direction variability model was found to relatively closely predict the trend of the yaw offsets achieved by the wake steering controller. Specifically, because



of imperfect yaw tracking in variable wind conditions, the achieved yaw offsets were found to be lower than the target offsets in the intended wake steering region, with undesired yaw offsets persisting outside of the intended sector. As reflected by the measured impact on energy production, these unintended yaw offsets appear to cause slight reductions in energy for wind directions outside of the intended wake steering sector. Overall, the wind direction variability model offers a simple way of

accounting for unintentional yaw misalignment when optimizing robust wake steering control strategies.

Despite the increase in energy production observed in this study, there are several opportunities to improve the performance and field validation of wake steering. First, a conservative yaw offset schedule was employed in this experiment to limit the impact of yaw misalignment on structural loads. After performing a detailed load assessment for the specific wind turbine used, the effectiveness of wake steering could be increased by allowing larger yaw offsets for a wider range of wind speeds in addition

to leveraging both positive and negative yaw offsets. Further, whereas this experiment showed that the effectiveness of wake steering depends on wind speed, the energy gains achieved through wake steering strongly depend on atmospheric stability as well, as shown by Fleming et al. (2019, 2020). Thus, if relevant measurements are available, yaw offsets could be optimized and scheduled as a function of stability—or other variables related to stability such as turbulence intensity, as described by Doekemeijer et al. (2021)—in addition to wind speed and direction. Additionally, whereas we used an indirect wake steering

control strategy based on modifying the input to the wind turbine's existing yaw controller, more advanced controllers, such as those discussed in Section 1, could improve performance by directly controlling the yaw position and responding more quickly to changing wind conditions. Opportunities also exist to increase the accuracy of the inputs to the wake steering controller. For example, the consensus control strategy described by Annoni et al. (2019) uses information sharing between neighboring wind turbines to improve local wind direction estimates. Finally, the energy gains estimated in this study are accompanied by

a significant amount of uncertainty. In addition to extending the duration of wake steering experiments, we expect uncertainty can be greatly reduced by increasing the number of wind turbines used to validate the overall impact of wake steering. ENGIE and NREL aim to incorporate some of these improvements by collaborating on a larger-scale wake steering campaign as part of the upcoming AWAKEN experiment in the United States (Moriarty et al., 2020).

**Appendix A: Wind Speed Dependence of Energy Ratios for Combined Downstream and Upstream Turbines**

*Code availability.* The FLORIS code used to model wake steering performance and calculate the energy ratios in this paper is available at https://github.com/NREL/floris (NREL, 2021).

*Author contributions.* NG, TD, and PF envisioned the wake steering experiment, which was designed by TD, PF, and ES. ES, PF, EG, and TD analyzed the data and interpreted the results. ES performed the modeling steps, with significant contributions from EG. NG supervised and managed the project for ENGIE. LA contributed immensely to the implementation of the wake steering controller and data recording

**Figure A1.** Energy ratios for the baseline and the controlled periods for turbines SMV5 and SMV6 combined, binned by wind speed. Energy ratios are derived from field observations as well as from FLORIS calculations using observed yaw offsets. Shaded regions indicate the 95% confidence intervals of the energy ratios for individual wind direction bins. The gray dashed lines encompass the intended wake steering sector.

hardware. TD organized and monitored the field experiment. ES prepared the manuscript, with major contributions from TD and input from all co-authors.





*Competing interests.* The authors declare that they have no conflict of interest.

*Acknowledgements.* This work was authored in part by the National Renewable Energy Laboratory, operated by Alliance for Sustainable Energy, LLC, for the U.S. Department of Energy (DOE) under Contract No. DE-AC36-08GO28308. Funding provided by the U.S. Depart-
ment of Energy Office of Energy Efficiency and Renewable Energy Wind Energy Technologies Office. The views expressed in the article do not necessarily represent the views of the DOE or the U.S. Government. The U.S. Government retains and the publisher, by accepting the article for publication, acknowledges that the U.S. Government retains a nonexclusive, paid-up, irrevocable, worldwide license to publish or reproduce the published form of this work, or allow others to do so, for U.S. Government purposes.

ENGIE Green's contribution to this work was partly funded by the French national project SMARTEOLE (grant no. ANR-14-CE05-0034).
The authors thank all the operation-and-maintenance teams of ENGIE Green for their help in setting up the field test, in particular, Aurélien Belland, Jimmy Mical, and Johan Joyeux. Thanks also to Nathanaël Wybou from ENGIE Laborelec for his great help in designing and monitoring the control box system. Last, the authors thank Robb Wallen from NREL for his advice on the control box software.



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
