# Peer review of "Results from a Wake Steering Experiment at a Commercial Wind Plant: Investigating the Wind Speed Dependence of Wake Steering Performance"

_Wind Energy Science, 2021_

## Author Comment (AC1)

**Author Response to Reviewer 1**

Dear Reviewer,

Thank you for your review of this manuscript and for your interest in the topic. We have revised the manuscript based on your and the other reviewer's comments. You will find our responses to your comments below.

> Useful paper, very well presented. I have only minor comments.

We appreciate your interest in this research. We found your comments to be very insightful and believe that they have made the manuscript considerably stronger.

> Fig 6: The target offsets are changed stepwise as the wind speed increases. As the shape is always the same, why not allow a continuous change in scaling factor with wind speed?

This was done because the wake steering controller was implemented using a two-dimensional lookup table (wind direction and wind speed) rather than using calculations in real time. 1 m/s step sizes were chosen for simplicity. However, because of the relatively slow yaw controller dynamics (changing yaw position on the order of every few minutes as opposed to seconds), we were not concerned with being too detailed with the offsets and trying to respond to small changes in wind speed. However, as you suggested, continuously varying the target offsets as a function of wind speed is a valid approach too and could have advantages in terms of power gain by allowing larger offsets for certain wind speeds (loads permitting).

To address the reason for a stepwise wind speed dependence for the target offsets, we added the following sentence to the end of the paragraph discussing Fig. 6: "Note that the target yaw offsets are binned by wind speed in steps of 1 m/s rather than specified as a continuous function of wind speed to enable a simple lookup table implementation."

> Line 196 says, "the most relevant feature for the two-turbine scenario investigated here is "yaw-added recovery,"" but there doesn't seem to be any further mention of this effect in the paper.

We meant to explain that the main feature of the GCH model, which is secondary steering, isn't particularly relevant for this work, since it applies to wake steering for rows of at least 3 turbines. But yaw-added recovery is relevant even for two-turbine wake steering scenarios. However, we appreciate the feedback that the original phrase seemed to imply that yaw-added recovery would be discussed more in the paper. Instead, we have replaced the sentence with the following standalone high-level overview of the GCH model:

"The curl-specific elements of the GCH model include secondary steering—whereby the vortices created by a misaligned wind turbine deflect the wakes of downstream turbines with which they interact—and yaw-added recovery, in which the vortices increase wake recovery through mixing with higher velocity flow."

> Line 202: "The FLORIS model is tuned ... by adjusting the turbulence intensity input" - What does this imply? If you use the measured turbulence intensity, does the model not fit so well?

There are a few reasons we don't use the measured turbulence intensity as an input to the FLORIS model. First, the available reference measurements of turbulence intensity are likely unreliable. Turbulence measured by the nacelle anemometers will be impacted by the rotor wakes. Additionally, the two lidars provide turbulence measurements. But because of volume averaging and cross-contamination of the velocity components within the line-of-sight measurements at the different beam locations, turbulence intensity measurements from lidars are typically biased (compared to a cup or sonic anemometer). Lastly, the model describing the relationship between ambient turbulence intensity and wake recovery and expansion (based on Niayifar and Porté-Agel, 2015) was developed using LES simulations. To our knowledge, it has not been validated using field data. Therefore, we prefer to treat turbulence intensity as a tuning parameter. However, we believe that future work is needed to carefully validate the turbulence model using field data. (The model parameters would likely depend on the time resolution of the field data as well. We expect 1-minute SCADA data would yield a different set of model parameters than 10-minute SCADA data because of the effects of wind direction variability and wake meandering within the averaging period.)

We added the following sentence to the second paragraph of Section 3.1 to explain our reasoning:

"We treat turbulence intensity as a tuning parameter rather than using the measured turbulence intensity as an input to FLORIS because 1) the turbulence intensity measurements provided by the ground-based and nacelle lidars do not represent traditional turbulence measurements (e.g., from a cup or sonic anemometer) because of volume averaging and line-of-sight measurement limitations (Kelberlau and Mann, 2020), and 2) further work is required to validate the relationship between turbulence intensity and wake deficits (Niayifar and Porté-Agel, 2015) that is used in the GCH model."

> Line 266: "it helps ensure that none of the turbines are unavailable or curtailed" - were there no flags available to indicate such turbine states?

We originally removed data when any of the test or reference turbines were producing less than 10 kW as a conservative way to make sure all of the turbines were operating normally. We had used the same approach in previous field experiments when we did not have access to turbine status codes and repeated it here. However, we do have status codes for the turbines and after considering this comment, we decided to use them to filter out periods with derating,

curtailment, or other forced downtime related to maintenance or faults, etc. Because of this step, we no longer remove periods based on whether any of the turbines are producing less than 10 kW. Some of the main advantages of this new approach are that we should be able to see more of the benefit of wake steering in preventing the downstream turbine from shutting down due to low wind speeds in the wake (because we no longer remove samples when the downstream turbine is shut down as a result of below cut-in wind speeds). Additionally, the energy gains and reductions in wake losses that we present more accurately reflect the true potential of wake steering at low wind speeds, since we are no longer selecting for periods when all turbines happen to be operating, which reduces the relative frequency of low wind speed periods in the overall data set.

But for the analyses of the achieved yaw offsets in Section 5 and the impact of yaw misalignment on power production in Section 6, we still remove periods when the controlled turbine is generating less than 10 kW. This ensures that the turbine is operating and therefore capable of reacting to yaw offset commands.

We also filter out data when the reference wind speed is below 4 m/s because little power production is expected for these wind speeds (and very few data were collected below 4 m/s).

Note that because we are removing fewer data samples now, most of the results have changed slightly, including Figures 7, 8, 9, 11-23, and A1, and the estimated wake loss reduction values reported in Sections 7.1 and 7.3. The most significant changes are the long-term corrected wake loss reduction (9.0% instead of the previous estimate of 9.8%), the best-fit lines in Fig. 15 relating the nacelle wind vane-measured yaw offsets to the nacelle lidar-measured offsets, and the cosine exponents used to model the power loss from yaw misalignment in Fig. 16 (For wind speeds from 4-8 m/s, the exponents are close to 2.25 instead of the previous 2.5). For the results in Sections 5.3 and 6, another reason the results changed slightly is because data from more wind directions (up to 270 degrees) are now included. We always intended to use these wind directions, but we found a bug that was discarding some of the data. However, the general trends and conclusions discussed in the original draft are not affected by the new values.

We have clarified our new filtering approach in Section 4.1:

"To improve the likelihood that observed differences in power production for the baseline and the controlled periods are caused by wake steering rather than abnormal turbine operation, the data are filtered using the following steps. First, periods for which the reference wind speed (which will be discussed in Section 4.2) is less than 4 m/s are removed because relatively few samples were collected for these wind speeds (see Fig. 7) and little energy production is expected during these conditions (as shown in Fig. 9). Periods with known derating, curtailment, or other forced downtime (aside from periods with wind speeds below the cut-in speed) are then removed from the data set by examining the wind turbine status codes. Next, any remaining periods with anomalous power production for any of the test or reference

turbines are removed using power curve filtering functions available in NREL's OpenOA software…

…

When investigating the yaw offsets achieved by the controlled wind turbine, SMV6, in Section 5 or the impact of yaw misalignment on the power production of SMV6 in Section 6, additional filtering steps are performed. First, samples for which SMV6 is generating less than 10 kW are removed. This step ensures that the turbine is operating and capable of responding to yaw offset commands. Next, because the analyses in Sections 5 and 6 rely on the WindCube Nacelle lidar measurements, the data are additionally filtered to remove 1-minute samples in which…"
* * *
Line 268: What are "the power curve filtering functions available in NREL's OpenOA software"? Some indications would be useful to help reassure that no biases are introduced by any of the processing.
* * *
We have added the following description in the text:

"Next, any remaining periods with anomalous power production for any of the test or reference turbines are removed using power curve filtering functions available in NREL's OpenOA software (Perr-Sauer et al., 2021) as follows (see Fig. 9 for context):

–Samples for which the nacelle wind speed measurement is greater than 6 m/s and power is less than 1% of rated power or greater than 101.5% of rated power are removed.

–Samples for which the nacelle wind speed measurement is greater than 14 m/s and power is less than 91.5% of rated power are removed (note that the manufacturer-specified rated wind speed is 14.5 m/s).

–The data are grouped by power into 50 bins with bin edges evenly distributed between 1% and 91.5% of rated power. Within each power bin, samples for which the difference between the nacelle measured wind speed and the median wind speed exceeds two standard deviations are removed. The threshold is increased to three standard deviations for SMV6 to account for increased power variability from intentional yaw misalignment."
* * *
Line 286: "identifying the wind direction where the ratio between the mean power produced by SMV5 and SMV6 reaches a minimum" - presumably only with wake steering off, and making the (quite reasonable) assumption that the wake deflection at zero yaw is small.
* * *
Yes, we limited the comparison to baseline periods when wake steering was not active and assumed that the natural wake deflection with zero yaw misalignment is close to zero. We clarified this in the modified text below:

"The reference wind direction is calibrated to true north by first identifying the measured wind direction where the ratio between the mean power produced by SMV5 and SMV6 during baseline periods reaches a minimum, representing the direction where the wake losses suffered by SMV5 reach their peak. Assuming negligible wake deflection relative to the true wind direction during baseline operation, the offset between this observed wind direction and the known direction of alignment between SMV6 and SMV5 is then subtracted from the reference wind direction."

> Line 298 and other places: The term 'transfer function' usually applies in the frequency domain, but in this case I assume it's just a multiplier which is a function of one or more inputs. For clarity, it would be good to state this somewhere and specify what the inputs are.

We are most familiar with the term "transfer function" from an electrical engineering perspective, where it is a function of frequency. But "transfer function" is commonly used in the wind industry to describe a mapping from a measured wind speed to a freestream-equivalent wind speed (e.g., the nacelle transfer function defined in the IEC 61400-12-2 standard). Therefore, we wanted to use the term to describe similar functions applied in this work. But we have now clarified the transfer function definitions and how they are applied to the measurements through the following modifications to the last two paragraphs of Section 4.2 (note that transfer functions are described in Section 5.3 as well, but we feel that the first three sentences of the second paragraph of Section 5.3 adequately define the transfer function for that section):

"Similarly, the reference wind speed is based on the mean wind speed measured by SMV1, SMV2, SMV3, and SMV7 using nacelle anemometry. We then apply additional steps to estimate the freestream equivalent wind speed encountered by the test turbines. First, to account for sensor bias, wake effects, and the impact of terrain and surface roughness on local wind conditions (e.g., the forest south of SMV7), a wind direction and wind speed-dependent transfer function (i.e., a multiplier that is a function of wind direction and wind speed) is applied to the reference wind speeds to remove any bias from the wind speeds measured by SMV6 during baseline operation. This transfer function is estimated as the ratio between the mean wind speed measured by SMV6 and the mean uncorrected reference wind speed for the baseline periods, binned by the reference wind direction (in overlapping 3° bins) and uncorrected reference wind speed (in 1-m/s bins). Next, a nacelle transfer function (a wind speed-dependent multiplier in this case) is applied to estimate the freestream wind speed from the nacelle anemometer-based reference wind speed. The nacelle transfer function is calculated as the ratio between the mean wind speed measured by the WindCube Nacelle lidar at a range of 150 m (1.8D) upstream of the rotor and the mean reference wind speed— considering only periods with baseline control and wind directions with freestream inflow— binned by the reference wind speed in 1-m/s bins.

Last, the reference power is formed by averaging the power production of SMV1, SMV2, SMV3, and SMV7. Following an approach similar to the reference wind speed derivation, a transfer function is applied to the average power produced by the four reference wind turbines to remove any bias from the power generated by SMV6 during baseline operation (e.g., caused by differences in turbine performance, wake effects, or the impact of local terrain and surface roughness). Again, this transfer function is a wind direction and wind speed-dependent multiplier that is estimated by dividing the data into overlapping 3° reference wind direction bins as well as 1-m/s reference wind speed bins, then calculating the ratio between the mean power produced by SMV6 and the mean uncorrected reference power for periods with baseline control."

> Figure 14: linear regression - there is a slight but distinct curve - would it make sense to fit a quadratic?

We agree that Figs. 14 and 15 show a slightly nonlinear relationship between the vane-measured and true yaw offsets and believe that this is worth discussing in the paper. However, we applied a simple linear fit to the curves to reveal the wind vane bias as well as how much the vane generally under- or overestimates the magnitude of the true yaw offsets. Additionally, a higher order curve fit risks over-fitting the data. To explain our reasoning for using a linear approximation while also acknowledging that the true relationship might be more complex, we added the following sentence to the second paragraph of Section 5.3:

"Although the actual relationship between the vane-measured and true offsets could be nonlinear, we use a simple linear approximation to reveal 1) the wind vane bias when zero yaw misalignment is reported, and 2) the overall degree to which the wind vane tends to underestimate or overestimate the magnitude of the yaw misalignments."

and the following sentence to the last paragraph of Section 5.3:

"Note that because of the somewhat nonlinear relationship between the vane-measured and true yaw offsets shown in Figs. 14 and 15, more sophisticated transfer functions (e.g., higher order polynomial functions) might be more appropriate than the linear approximations presented here. Further, the amount of wind vane bias…"

> Line 410: "blade element momentum theory predicts Pp = 3." This is only true if a skewed wake correction is not used. It is recommended to use a skewed wake correction in blade element theory for better prediction of performance in yaw.

Thank you for pointing this out. We have modified the sentence to "But as discussed by Liew et al. (2020), a value of $p_P$ = 3 is predicted by blade element momentum theory, albeit without a skewed wake correction."

Line 411: "they are expected to agree in below-rated wind speeds" Above rated, equation (3) is definitely wrong, so is there a reason not to use equation (1)? The processing is only slightly less straightforward, effectively shifting the power curve to the right. The offset (alpha) can also be used in equation (1).

We agree that above rated, yaw misalignment should have negligible impact on power. But this could be captured by a wind speed-dependent cosine exponent, which is what we are calculating in Fig. 16 (the cosine exponent should be 0 above rated). Further, we plot the $p_P$ cosine exponents from Eq. 3 because this is the definition of the cosine exponent that tends to be used more often in the wind community/literature. Therefore, we see value in providing these estimates for comparison with other published values. Additionally, we believe that highlighting how the cosine exponent gradually decreases as wind speed increases (not just one value below rated and 0 above rated) is interesting to the wind community. Finally, we feel that plotting the normalized power as a function of yaw offset for different wind speeds is a compelling way to communicate how the impact of yaw misalignment on power decreases as wind speed increases toward rated.

However, you raise a good point that since Eq. 1 is assumed to be a more physically realistic formula (if we want to find a single cosine exponent independent of wind speed) and is the method used in FLORIS, it makes sense to estimate the exponent $p_V$ from the field measurements. Therefore, we estimated these values and provided them in Table 1, where they are compared to the $p_P$ values from Fig. 16.

We don't think it is quite as straightforward to estimate $p_V$ from field measurements as it is for $p_P$, however. The method we developed uses the mean absolute test and reference power values for each yaw offset bin and finds the equivalent wind speeds that would have produced those power values using the inverse of the measured power curve shown in Fig. 9. The method is tricky near rated wind speed because the inverse of the power curve yields undefined wind speeds once the power reaches rated power. Therefore, we use the inverse of the power curve for wind speeds up to 14.5 m/s and extrapolate when the measured power is greater than the power from the power curve at 14.5 m/s.

Once we estimate the equivalent test and reference wind speeds for each yaw offset, we fit the same cosine function as used previously to estimate $p_v$ from the ratios between the test and reference wind speeds as a function of yaw offset.

In addition to adding Table 1, to describe the method for estimating $p_v$ and discuss the findings, we modified the first paragraph of Section 6 and added two paragraphs (the 2nd and 3rd paragraphs from the end of the section) as follows (although these are the main additions, the full changes to the section can be found in the marked up revised draft):

"Understanding the relationship between yaw misalignment and power production is paramount to the design of optimal wake steering strategies. In this section, we use measurements from the WindCube Nacelle lidar to estimate the impact of yaw misalignment

on power production. Although later in the section we estimate the cosine exponent, $p_v$, used by FLORIS to model the power loss from yaw misalignment via the effective wind speed (see Eq. 1), we first use the more traditional method…

…

Next, we estimate the cosine exponent, $p_v$, used by FLORIS to describe the impact of yaw misalignment on the effective wind speed (see Eq. 1) by slightly modifying the method used to estimate $p_P$. Instead of finding the best-fit cosine exponent using the ratios between the measured power, $P$, and reference power, $P_0$, binned by yaw misalignment, γ, we fit the function $\cos(\gamma-\alpha)^{pv/3}$ (based on Eq. 1) to the ratios between the effective wind speeds that correspond to $P$ and $P_0$ as a function of γ. The effective wind speeds are estimated by finding the wind speeds that map to $P$ and $P_0$ using the measured power curve shown in Fig. 9 (note that this method is unreliable when the measured power is greater than or equal to rated power).

Estimates of the $p_v$ cosine exponents are compared to the $p_P$ estimates from Fig. 16 as a function of wind speed in Table 1. As anticipated, compared to $p_P$, the $p_v$ exponents remain relatively constant (between 1.4 and 2.1) across all wind speed bins because the impact of yaw misalignment on the effective wind speed described by Eq. 1 is expected to be roughly independent of wind speed. Note that for wind speeds between 4–8 m/s, where the two different cosine exponents are expected to closely agree, we find that the estimated $p_v$ values are lower than the corresponding $p_P$ estimates; however, as indicated by the 95% confidence intervals in Table 1, the estimation uncertainty is high for both variables. Despite the variations in the estimated $p_v$ values for different wind speed bins, the *relative* stability of the $p_v$ estimates as a function of wind speed compared to the corresponding $p_P$ values justifies the use of Eq. 1 to model the impact of yaw misalignment on power production in FLORIS. Finally, we note that the mean value of the $p_v$ estimates listed in Table 1 of 1.68 is close to the value of 1.6 used in the FLORIS model for this study.”

| Line 446: "depends on the atmospheric boundary layer as well as the turbine's control system" It also depends on the rotor aerodynamics. |
| --- |

Good point, and the rotor aerodynamic properties are an important part of the model presented by Howland et al. (2020), which is cited in this paragraph. We have modified the following sentences:

"Last, although Fig. 16 reveals the impact of yaw misalignment on power production for a specific wind turbine, recent research suggests that the relationship between yaw misalignment and power depends on the atmospheric boundary layer as well as the turbine's aerodynamic properties and control system. Using rotor airfoil properties, Howland et al. (2020) show how the power production of a misaligned wind turbine depends on the wind shear and veer profiles interacting with the rotor, which can introduce asymmetry in the relationship between yaw misalignment and power.”

Line 582: Presumably the measured mean values in each bin are used. For clarity it might be worth spelling out precisely how this confidence interval is obtained for the long-term weighted results?

To calculate the long-term corrected energy ratios and wake loss reduction values, we bin the data into 2°-by-1-m/s wind direction/wind speed bins and find the mean power within each bin for the test and reference turbines. The long-term wind rose frequencies are determined for each bin by calculating the fraction of the total long-term data that is within each fixed 2°-by-1-m/s wind direction/wind speed bin.

The confidence interval for the long-term corrected wake loss reduction is estimated using bootstrapping by randomly resampling the data (containing all wind directions and wind speeds) for baseline and controlled periods independently, as explained in Section 4.3. However, the long-term wind condition frequencies are kept fixed at the values determined from the long-term wind rose in Fig. 2 – no resampling of the long-term wind condition data was performed.

We have added some clarification on the bin sizes used for the energy ratio and wake loss reduction calculations as well as how the long-term frequencies are determined for each bin throughout Section 7 as follows:

First paragraph of Section 7: "…the weighting factor $w_i$ is defined as the total number of samples in wind speed bin $i$ for the baseline and the controlled periods combined, and $N_{WS}$ indicates the number of 1-m/s-wide wind speed bins used in the calculation."

Last paragraph of Section 7.1: "Wake losses are calculated for the baseline and the controlled periods separately by first binning the difference between the reference power and the average power produced by SMV5 and SMV6 by wind direction (in 2° bins) and wind speed (in 1-m/s bins)."

First paragraph of Section 7.3: "The long-term corrected energy ratio calculation requires only a slight modification to the energy ratio definition in Eq. 4; instead of weighting the mean power for the test and reference turbines in a particular 2° × 1-m/s wind direction and wind speed bin by the total number of samples measured in that bin, we weight the mean power by the long-term frequency of occurrence of the wind conditions within the bin."

Last paragraph of Section 7.3: "But similar to the long-term corrected energy ratio method, we modify the procedure for calculating the wake losses by weighting the difference between the mean power of the reference and test turbines in each 2° × 1-m/s wind direction and wind speed bin by the long-term frequencies of occurrence of the wind conditions within the bin rather than by the frequencies observed during the experiment period."

To clarify how the confidence intervals are calculated for the long-term corrected wake loss reduction, we have added the following sentence to the last paragraph of Section 7.3:

"Note that this confidence interval is estimated through bootstrapping by randomly resampling the data used to calculate the mean power values, as explained in Section 4.3; however, the long-term wind condition frequencies are fixed at the values determined from the long-term wind rose shown in Fig. 2."

> Line 606: "12–14-m/s wind speed bin—just below the turbines' rated wind speed" - That depends how you define rated wind speed. If you define it as the wind speed at which rated power is reached in steady wind, I would estimate the rated wind speed at around 11.5 m/s from visual inspection of the (turbulent) power curve.  The higher the turbulence, the higher the wind speed at which the 10-minute average power "reaches" rated (inverted commas because in theory it never quite reaches rated power if there's any turbulence.)

It's true that the rated wind speed is hard to define in turbulent wind conditions because of power production for a range of wind speeds contributing to each measured data point (assuming an averaging time of at least 10 s or so). We agree that the rated wind speed in steady conditions is likely in the range of 11-13 m/s. However, we say "just below the turbines' rated wind speed" because the manufacturer's advertised rated wind speed is 14.5 m/s and manufacturer power curves are typically provided for a specific turbulence class.

We have changed this phrase to "just below the turbines' manufacturer-specified rated wind speed of 14.5 m/s"

> Line 608: "wake loss reduction from wake steering increases to 9.8% for the wind directions analyzed" - Perhaps it's obvious, but taken over all wind directions, the improvement would be smaller - but, equally obviously, if wake steering were applied to the whole farm, not just one turbine, there would be more to gain.

We use the metric of percentage of wake losses reduced by wake steering rather than percentage energy gain or percentage AEP gain partially because it avoids ambiguities regarding which wind directions to include in the calculation. Specifically, the wake loss reduction from wake steering for the controlled turbine SMV6 waking SMV5 should be independent of the wind direction sector used, as long as the sector encompasses wind directions where SMV5 is waked by SMV6. For example, if we calculated the wake loss reduction for the wind direction sector from 195° to 330° instead of the current 195° to 240°, we would expect the wake loss reduction to remain at roughly 9.0% (the new value in the revised version of the manuscript), since there would be no additional wake losses for these two turbines (and therefore wake loss reduction opportunities) for wind directions from 240° to 330°.

But we agree that if we considered all wind directions from 0 to 360°, there would be additional wake losses experienced by SMV5 and SMV6 from other turbines, and the total wake loss

reduction for SMV5 and SMV6 would depend on whether SMV5 is also controlled for northeasterly flow. Similarly, the wake loss reduction for the entire plant would depend on which wind directions are included and which turbines are controlled.

We have now commented on how the wake loss reduction could be affected by which wind directions are considered and which turbines are controlled in the second paragraph of Section 8:

"Note that the wake loss reduction values estimated here represent the reduction in wake losses caused by the controlled wind turbine, SMV6, waking the downstream turbine, SMV5; the potential reduction in wake losses from wake steering for the entire wind plant, across all wind directions, will depend on several factors (e.g., the particular set of wind turbines that are controlled and the yaw offset schedules that are used)."

Line 615 "almost no impact on power production from yaw misalignment was detected" - This is expected because the turbine is actually above the 'steady' rated wind speed - see Line 606 comment.

We agree that this is not surprising because the power curve is relatively flat in this region. We have now commented on this in the text:

"As wind speed increases to 12–14 m/s—just below the official rated wind speed for the turbine investigated—almost no impact on power production from yaw misalignment was detected; this is expected because changes in the effective wind speed from yaw misalignment result in relatively small reductions in power for this wind speed bin (see Fig. 9)."

**Author Response to Reviewer 2**

Dear Reviewer,

Thank you for your review of this manuscript and for your interest in this topic. We have revised the manuscript based on your and the other reviewer's comments. Please find our responses to your comments below.

> Dear authors,
> thank you very much for the well prepared article.
>
> You describe and analyze the results of a wake steering experiment and put a focus on the wind speed dependent performance of wake steering.
>
> From my understanding the topic is very relevant in the current state of the technology because it gives results of wake steering field testing under realistic conditions. This helps to better understand the shortcomings and rate the expected performances of investigations.
>
> The analyzes are very complete and detailed and there are only a minor point I would like to address besides the points the first reviewer has already risen.

Thank you, we appreciate your interest in this research.

We'd like to point out that most of the results have changed slightly, including Figures 7, 8, 9, 11-23, and A1, and the estimated wake loss reduction values reported in Sections 7.1 and 7.3. This is primarily because we are filtering out fewer data samples now, as explained in the response to the first reviewer's 4th comment. The most significant changes are the long-term corrected wake loss reduction (9.0% instead of the previous estimate of 9.8%), the best-fit lines in Fig. 15 relating the nacelle wind vane-measured yaw offsets to the nacelle lidar-measured offsets, and the cosine exponents used to model the power loss from yaw misalignment in Fig. 16 (For wind speeds from 4-8 m/s, the exponents are close to 2.25 instead of the previous 2.5). For the results in Sections 5.3 and 6, another reason the results changed slightly is because data from more wind directions (up to 270 degrees) are included. We always intended to use these wind directions, but we found a bug that was discarding some of the data. However, the general trends and conclusions discussed in the original draft are not affected by the new values.

> The naming convention "ideal", "predicted", "achieved" is sometimes difficult to understand/distinguish. To my understanding the ideal are the one from FLORIS without wind direction variability model, the predicted are the FLORIS with wind direction variability, and the achieved are the measured ones.
>
> Maybe you can find a better terminology which is more self-explaining.

You are correct. The ideal offsets are the target yaw offsets determined from the yaw offset schedule, the predicted offsets are determined from the wind direction variability model, and the achieved offsets are just the measured offsets. We appreciate the feedback that the naming convention is unclear and have arrived at "measured offsets", "ideal offsets", and "expected offsets" as replacements. We also use these terms more consistently throughout the paper (e.g., instead of saying "predicted" sometimes and "predicted achieved" other times). Further, for the legend names in Section 7, we changed "FLORIS ideal gain" to "FLORIS Gain, Ideal Offsets", etc., which we feel is a more self-explanatory name. However, to clarify the naming convention further, we have added a paragraph to the end of Section 3 formally define these terms.

The following paragraph is now included at the end of Section 3:

"When analyzing the yaw offsets during the wake steering experiment in Section 5 as well as the change in energy from wake steering predicted by FLORIS in Section 7, the following nomenclature will be used to distinguish different methods for determining the yaw offsets:

–Measured offsets: The yaw offsets measured using SMV6's nacelle wind vane or the WindCube Nacelle lidar.

–Ideal offsets: The target yaw offsets determined from the yaw offset schedule shown in Fig. 6 as a function of the reference wind direction and wind speed (which will be discussed in Section 4.2).

–Expected offsets: The yaw offset distributions predicted by the wind direction variability model presented in this section as a function of the reference wind direction and wind speed."

We have also modified the first paragraph of Section 7.1 to remind the reader of these definitions when presenting energy gain predictions using FLORIS:

"The overall energy ratios and the change in energy ratio for the baseline and the wake steering control periods for the downstream turbine, SMV5, are plotted in Fig. 17 as a function of wind direction, along with 95% confidence intervals. The measured energy ratios and the change in energy ratio with wake steering are compared to the same metrics based on FLORIS simulations using the three different FLORIS modeling assumptions discussed in Section 3.2. First, FLORIS estimates of power production are calculated for the observed distribution of wind directions, wind speeds, and yaw offsets measured using SMV6's nacelle wind vane (labeled "Measured Offsets"). Next, the ideal FLORIS estimates are calculated using the intended yaw offsets for SMV6 as a function of the observed wind direction and wind speed according to the yaw offset schedule shown in Fig. 6 (labeled "Ideal Offsets"). Last, the realistic expected energy ratios based on FLORIS are calculated by combining the ideal yaw offsets for SMV6 with the wind direction variability model discussed in Section 3.2 (labeled "Expected Offsets")."

Additionally, throughout the manuscript (text and figure captions), the nomenclature has been updated to reflect these definitions, more consistently using "Measured offsets", "Ideal offsets", and "Expected offsets" to describe how the yaw offsets were determined.